# Catch the Shadow: Automatic Shadow Variables Generation for Treatment Effect Estimation under Collider Bias

## Abstract

Collider bias, which comes from non-random sample selection caused by both treatments and outcomes, is a significant and challenging problem of treatment effect estimation. Previous studies show that treatment effects are identifiable if some shadow variables are available in the observational data. Shadow variables are assumed to be fully observed covariates independent of the sample selection mechanism after conditioning on the outcome and other observed covariates. However, finding a well-defined shadow variable is often not an easier task than the task of dealing with collider bias itself in real-world scenarios. Therefore, we propose a novel ShadowCatcher that automatically generates representations serving the role of shadow variables from the observed covariates. Specifically, during the generation process, we impose conditional independence constraints on the learned representations to make them satisfy the assumptions of shadow variables. To further ensure that the generated representations are valid, we also use a tester to perform hypothesis testing and iteratively carry out the generation process until the generated representations pass the test. Using the generated representations, we propose a novel ShadowEstimator to estimate treatment effects under collider bias. Experimental results on both synthetic and real-world datasets demonstrate the effectiveness of our proposed ShadowCatcher and ShadowEstimator.

## 1 Introduction

Causal inference is a powerful statistical modeling tool for explanatory analysis, and a central problem in causal inference is treatment effects estimation. The gold standard approach for treatment effect estimation is to conduct Randomized Controlled Trials (RCTs), but RCTs can be expensive (Kohavi & Longbotham, 2011) and sometimes infeasible (Bottou et al., 2013). Therefore, developing practical approaches to estimate treatment effects from observational data is crucial for causal inference.

In observational studies, association does not imply causation, mainly due to the presence of spurious associations in the data. There are two primary sources of spurious associations: confounding bias and collider bias (Hernán & Robins, 2020). Most of the previous works focused on confounding bias that results from common causes of treatments and outcomes (Bang & Robins, 2005; Shalit et al., 2017; Louizos et al., 2017; Wager & Athey, 2018) while ignored collider bias which comes from non-random sample selection caused by both treatments and outcomes.

We use causal diagrams in Figure 1 to further illustrate the two biases, where $\mathbf{X}$ denotes the observed covariates, $T$ denotes the treatment variable, $Y$ denotes the outcome variable, and $S$ denotes the sample selection indicator. Confounding bias results from common causes of treatment and outcome (Greenland, 2003; Guo et al., 2020). As shown in Figure 1(a), there are two sources of association between $T$ and $Y$: the path $T \rightarrow Y$ that represents the treatment effect of $T$ on $Y$, and the path $T \leftarrow \mathbf{X} \rightarrow Y$ between $T$ and $Y$ that includes the common cause $\mathbf{X}$, which introduces spurious associations into the observational data. Collider bias is a particular case of sample selection bias[1] that results from conditioning on a common effect of $T$ and $Y$ (Hernán & Robins, 2020). As shown in Figure 1(b), except for the path $T \rightarrow Y$, the other source of association between $T$ and $Y$ is from

---

[1]Sample selection bias results from non-random sample selection conditioned on $S$ caused by certain variables in data, while collider bias is the particular case that $T$ and $Y$ both cause $S$.

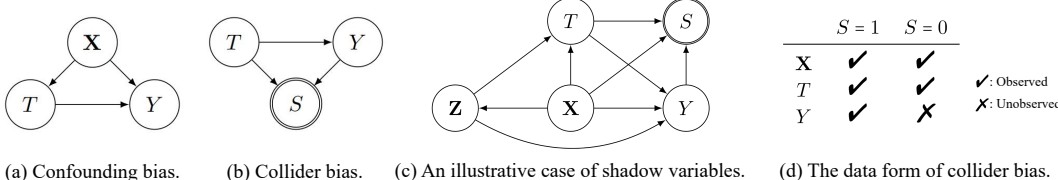

(a) Confounding bias.  (b) Collider bias.  (c) An illustrative case of shadow variables.  (d) The data form of collider bias.

Figure 1: Different kinds of biases represented by causal diagrams.

the open path $T \rightarrow S \leftarrow Y$. It links $T$ and $Y$ through their conditioned on common effect $S$, which introduces spurious associations. As shown in Figure 1(d), an analysis conditioned on $S$ will cause collider bias, i.e., we can only observe the outcome of those selected units ($S = 1$), and the values of $Y$ are missing for those unselected units ($S = 0$), leading to incorrect treatment effect estimation.

Previous studies show that treatment effects are unidentifiable under collider bias without further assumptions or prior knowledge. Fortunately, if some shadow variables are available in the observational data, it is still possible to identify treatment effects from observational data (Miao & Tchetgen Tchetgen, 2016). As shown in Figure 1(c), shadow variables $\mathbf{Z}$ are assumed to be fully observed covariates independent of the sample selection mechanism after conditioning on the outcome and other covariates, i.e., a valid shadow variable needs to simultaneously satisfy that $\mathbf{Z} \not\perp\!\!\!\perp Y \mid \mathbf{X}, T, S = 1$ and $\mathbf{Z} \perp\!\!\!\perp S \mid \mathbf{X}, T, Y$. For example, when studying the effect of students' mental health ($T$) on teachers' assessment ($Y$), collider bias occurs since teachers might not be willing to report their assessment of students with poor mental health. The teacher's response rate ($S$) may be related to their assessment of the student but is unlikely to be related to a separate parent's report after conditioning on the teacher's assessment and fully observed covariates; moreover, the parent's report is likely highly correlated with the teacher's. In this case, the parental assessment can be considered a shadow variable (Ibrahim et al., 2001). With the help of shadow variables, treatment effects can be identified and estimated (d'Haultfoeuille, 2010; Wang et al., 2014; Miao & Tchetgen Tchetgen, 2016).

However, finding a well-defined shadow variable requires domain-specific knowledge of experts and needs to be investigated on a case-by-case basis, which is often as challenging a task as the task of dealing with collider bias itself in real-world scenarios (Li et al., 2023). Therefore, we propose a novel method named **ShadowCatcher** that automatically generates representations from the observed covariates satisfying the assumptions of shadow variables, which can serve the role of shadow variables in the treatment effect estimation process and thus achieve the goal of solving collider bias without introducing more prior knowledge. Specifically, we iteratively generate shadow-variable representations by conditional independence constraints and test whether the generated representations satisfy the assumptions until the generated representations can pass the hypothesis test. Furthermore, we also propose a novel **ShadowEstimator** to estimate treatment effects under collider bias by leveraging the generated shadow variables representations. We conduct experiments on synthetic and real-world datasets, including ablation studies, and the results demonstrate the effectiveness of our proposed ShadowCatcher and ShadowEstimator.

The main contributions in this paper are as follows: (1) We study a practical and challenging problem of treatment effect estimation from observational data under collider bias. (2) We propose a novel ShadowCatcher that automatically generates representations serving the role of shadow variables from the observed covariates, which overcomes the common difficulty in finding valid shadow variables in real-world scenarios. (3) We propose a novel ShadowEstimator to estimate treatment effects using the generated shadow variable representations to address the collider bias in observational data. (4) Extensive experiments show that our proposed methods can practically generate shadow variable representations and address collider bias in treatment effect estimation.

## 2 RELATED WORK

Previous works on treatment effect estimation mainly focus on confounding bias in observational studies. Reweighting methods either use the inverse propensity score (Dehejia & Wahba, 2002) or learn a balancing weight from data (Hainmueller, 2012; Athey et al., 2018) to make $T$ and $\mathbf{X}$ of the reweighted samples independent. Balanced representation learning methods (Johansson et al.,

2016; Shalit et al., 2017; Greiner, 2020) learn representations of covariates so that the learned representations are independent of the treatment variable. Causal Forest (Wager & Athey, 2018) builds a large number of causal trees and then estimates heterogeneous treatment effects by taking an average of the outcomes from these causal trees. Generative methods (Louizos et al., 2017; Yoon et al., 2018) utilize generative models to generate counterfactual data. However, all the above methods suffer from sample selection bias because of the distribution shift problem.

To address sample selection bias, Heckman (1979) proposed a two-stage regression method with many extensions (Marchenko & Genton, 2012; Ding, 2014; Ogundimu & Hutton, 2016; Wiemann et al., 2022). Cole & Stuart (2010) proposed a sample reweighting method, which reweights the selected samples by estimating the inverse conditional probability of the sample selection as weights. Bareinboim et al. (2014); Bareinboim & Tian (2015) proposed the selection-backdoor adjustment approach by blocking the selection-backdoor paths. These methods can only solve selection bias caused by covariates and the treatment. However, these methods cannot solve collider bias, which is more likely to appear in real-world scenarios because $Y$ also causes $S$.

Fortunately, treatment effects are identifiable under collider bias if some shadow variables are available in the observational data (d'Haultfoeuille, 2010; Miao & Tchetgen Tchetgen, 2016). Shadow variables are assumed to satisfy that $\mathbf{Z} \not\perp Y \mid \mathbf{X}, T, S = 1$ and $\mathbf{Z} \perp\!\!\!\perp S \mid \mathbf{X}, T, Y$. With the help of shadow variables, various estimators, including regression-based (d'Haultfoeuille, 2010; Zhao & Shao, 2016), IPSW-based (Wang et al., 2014), and doubly-robust-based (Miao & Tchetgen Tchetgen, 2016) were proposed to solve collider bias. However, the accessibility of valid shadow variables itself is a strong assumption because finding a well-defined shadow variable requires domain-specific knowledge of experts and needs to be investigated on a case-by-case basis (Li et al., 2023). Therefore, our proposed method that automatically generates representations serving the role of shadow variables can effectively relax the assumptions of solving collider bias and has excellent application values.

## 3 PROBLEM AND ALGORITHM

### 3.1 PROBLEM FORMULATION

Suppose we have observational data $\mathcal{D} = \left\{ \mathbf{x}_i, t_i, y_i^{\text{obs}}, s_i \right\}_{i=1}^n$, where $n$ denotes the number of units. For the $i^{\text{th}}$ unit, we observe its treatment variable $t_i$, selection indicator $s_i$ that indicates whether the unit is selected into the sample, i.e., whether the value of the outcome can be observed, covariates $\mathbf{x}_i \in \mathbb{R}^{d \times 1}$, where $d$ denotes the dimension of the covariates, and observed outcome variable $y_i^{\text{obs}}$ remains the same value as $y_i$ when $s_i = 1$. For missing values, we label them as $s_i = 0$. Figure 1(d) illustrates the collected data form in the presence of collider bias.

In this paper, we focus on the case of binary treatment[2], i.e., $t_i \in \{0, 1\}$, where $t_i = 1$ denotes unit $i$ is treated, and $t_i = 0$ denotes otherwise. Under the potential outcome framework (Imbens & Rubin, 2015), we define the potential outcomes under treatment as $Y(1)$ and under control as $Y(0)$. With the observational data, our goal is to estimate the Conditional Average Treatment effect (CATE), which is defined as $\tau(\mathbf{x}) = \mathbb{E}[Y(1) - Y(0) \mid \mathbf{X} = \mathbf{x}]$. For a unit $i$ with $t_i$ in $\mathcal{D}$, only the factual outcome $Y(t_i)$ is available. Therefore, to make CATE identifiable, we make the following commonly used assumptions (Imbens & Rubin, 2015):

- **Stable Unit Treatment Value Assumption.** The distribution of the potential outcome of one unit is assumed to be independent of the treatment assignment of another unit.
- **Overlap Assumption.** A unit has a nonzero probability of being treated and being selected, $0 < \mathbb{P}(T = 1 \mid \mathbf{X} = \mathbf{x}) < 1$ and $0 < \mathbb{P}(S = 1 \mid \mathbf{X} = \mathbf{x}) < 1$.
- **Unconfoundedness Assumption.** The treatments are independent of the potential outcomes given the covariates, i.e., $Y(1), Y(0) \perp\!\!\!\perp T \mid \mathbf{X}$.

Based on the above assumptions, CATE can be estimated as:
$$\tau(\mathbf{x}) = \mathbb{E}[Y \mid \mathbf{X} = \mathbf{x}, T = 1] - \mathbb{E}[Y \mid \mathbf{X} = \mathbf{x}, T = 0]. \tag{1}$$
However, because the values of $Y$ are missing in $S = 0$ units caused by collider bias, we can only estimate the CATE of $S = 1$ samples, which differs from the true CATE of the entire data because

---

[2]In this paper, we mainly focus on how to generate shadow-variable representations to address collider bias. To make the proposed ShadowCatcher and ShadowEstimator process more concise, here we consider the binary treatment setting, but our proposed methods can also be effectively applied to continuous treatment settings.

$\mathbb{E}[Y \mid \mathbf{X} = \mathbf{x}, T = t, S = 1] \neq \mathbb{E}[Y \mid \mathbf{X} = \mathbf{x}, T = t]$. What is worse, since collider bias results in $Y(1), Y(0) \not\perp\!\!\!\perp T \mid \mathbf{X}, S = 1$, the unconfoundedness assumption is violated when conditioning on $S = 1$. It leads to a biased estimation using the observed samples, which means that the estimated CATE of the $S = 1$ samples even differs from the true CATE of only the $S = 1$ data. Therefore, it is necessary to develop approaches to solve collider bias for treatment effect estimation. Fortunately, studies show that treatment effects can be identifiable under collider bias if some shadow variables are available in the observational data (d'Haultfoeuille, 2010; Miao & Tchetgen Tchetgen, 2016).

### 3.2 Preliminaries of the shadow variable

Valid shadow variables $\mathbf{Z}$ are supposed to be fully observed covariates, i.e., the values of $\mathbf{Z}$ are observable in both $S = 0$ and $S = 1$ data like $\mathbf{X}$, and satisfy the following assumption:

**Assumption 1 (d'Haultfoeuille, 2010).** $\mathbf{Z} \not\perp\!\!\!\perp Y \mid \mathbf{X}, T, S = 1$ and $\mathbf{Z} \perp\!\!\!\perp S \mid \mathbf{X}, T, Y$.

As shown in Figure 1(c), Assumption 1 indicates that the shadow variable does not affect the sample selection mechanism after conditioning on the outcome and other observed covariates, and it is associated with the outcome given the covariates. This assumption is widely used in the literature of collider bias (d'Haultfoeuille, 2010; Wang et al., 2014; Miao & Tchetgen Tchetgen, 2016; Zhao & Shao, 2016; Li et al., 2023), and an illustrative example can be found in Section 1.

Throughout the paper, let $f(\cdot)$ denote the data distribution function. The key problem of collider bias is that the outcome values are missing in $S = 0$ data, which results in $f(Y \mid \mathbf{X}, \mathbf{Z}, T, S = 0)$ not available from the observed data. We can use the odds ratio function to encode the deviation between the distribution of $S = 1$ data and that of $S = 0$ data, which can be expressed as follows under Assumption 1 (Miao & Tchetgen Tchetgen, 2016)[3]:

$$\mathrm{OR}(\mathbf{X}, \mathbf{Z}, T, Y) = \mathrm{OR}(\mathbf{X}, T, Y) := \frac{f(S = 0 \mid \mathbf{X}, T, Y) \cdot f(S = 1 \mid \mathbf{X}, T, Y = 0)}{f(S = 0 \mid \mathbf{X}, T, Y = 0) \cdot f(S = 1 \mid \mathbf{X}, T, Y)}. \tag{2}$$

In Equation (2), $Y = 0$ is used as a reference value, and $\mathrm{OR}(\mathbf{X}, T, Y = 0) = 1$, which can be replaced by any other value within the support of $Y$. The odds ratio function measures the degree to which the $S = 0$ data differs from the $S = 1$ data and thus can be used to recover the unknown $f(Y \mid \mathbf{X}, \mathbf{Z}, T, S = 0)$ from the observed $f(Y \mid \mathbf{X}, \mathbf{Z}, T, S = 1)$ through the following proposition:

**Proposition 1 (Miao & Tchetgen Tchetgen, 2016).** Given Assumption 1, we have that

$$f(Y \mid \mathbf{X}, \mathbf{Z}, T, S = 0) = \frac{\mathrm{OR}(\mathbf{X}, T, Y) \cdot f(Y \mid \mathbf{X}, \mathbf{Z}, T, S = 1)}{\mathbb{E}[\widetilde{\mathrm{OR}}(\mathbf{X}, T, Y) \mid \mathbf{X}, \mathbf{Z}, T, S = 1]}, \tag{3}$$

$$\mathbb{E}[\widetilde{\mathrm{OR}}(\mathbf{X}, T, Y) \mid \mathbf{X}, \mathbf{Z}, T, S = 1] = \frac{f(\mathbf{Z} \mid \mathbf{X}, T, S = 0)}{f(\mathbf{Z} \mid \mathbf{X}, T, S = 1)}, \tag{4}$$

where $\widetilde{\mathrm{OR}}(\mathbf{X}, T, Y) = \mathrm{OR}(\mathbf{X}, T, Y)/\mathbb{E}[\mathrm{OR}(\mathbf{X}, T, Y) \mid \mathbf{X}, T, S = 1]$. Equation (3) shows that the key problem that $f(Y \mid \mathbf{X}, \mathbf{Z}, T, S = 0)$ is unidentifiable can be solved under Assumption 1 by integrating the odds ratio function with the $S = 1$ data distribution. Since $f(Y \mid \mathbf{X}, \mathbf{Z}, T, S = 1)$ can be obtained from the fully observed $S = 1$ samples, the only problem becomes the identification of the odds ratio function. Fortunately, With $f(\mathbf{Z} \mid \mathbf{X}, S = 0)$ and $f(\mathbf{Z} \mid \mathbf{X}, S = 1)$ obtained from the observed data, Equation (4) is a Fredholm integral equation of the first kind, with $\widetilde{\mathrm{OR}}(\mathbf{X}, T, Y)$ to be solved for. Because $\mathrm{OR}(\mathbf{X}, T, Y = 0) = 1$, we have the following result[4]

$$\mathrm{OR}(\mathbf{X}, T, Y) = \frac{\widetilde{\mathrm{OR}}(\mathbf{X}, T, Y)}{\widetilde{\mathrm{OR}}(\mathbf{X}, T, Y = 0)}. \tag{5}$$

Therefore, identification of $\mathrm{OR}(\mathbf{X}, T, Y)$ is equivalent to finding a unique solution to Equation (4), which is guaranteed by the following theorem:

**Theorem 1 (Miao & Tchetgen Tchetgen, 2016).** Under Assumption 1 and the completeness condition of $f(Y \mid \mathbf{X}, \mathbf{Z}, T, S = 1)$, Equation (4) has a unique solution. Thus $\mathrm{OR}(\mathbf{X}, T, Y)$ and $f(Y \mid \mathbf{X}, \mathbf{Z}, T)$ can be identified.

---

[3]See Appendix A.3.1 for more detailed explanation.

[4]See Appendix A.3.2 for more detailed explanation.

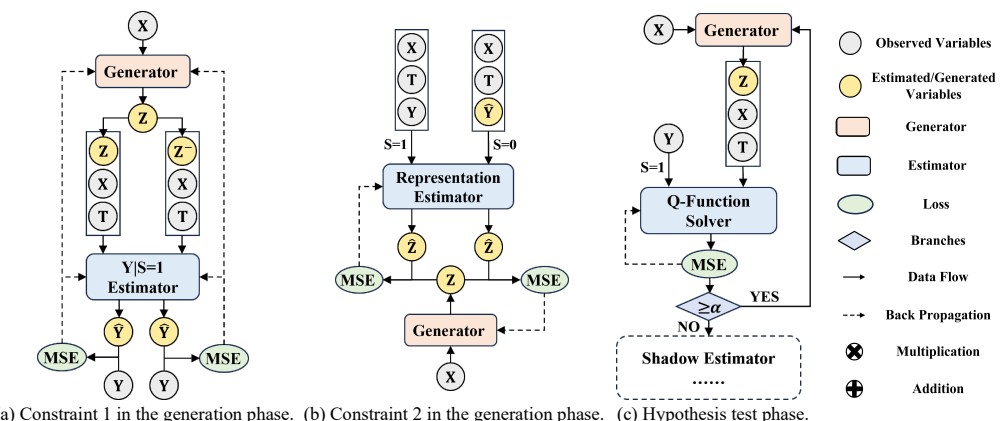

(a) Constraint 1 in the generation phase. (b) Constraint 2 in the generation phase. (c) Hypothesis test phase.

Figure 2: The flowchart of ShadowCatcher, including the generation phase and the test phase.

Based on the above theorem, collider bias can be solved with the help of shadow variables by firstly estimating $\text{OR}(\mathbf{X}, T, Y)$ through Equation (4) and (5), then recovering $f(Y \mid \mathbf{X}, \mathbf{Z}, T, S = 0)$ through Equation (3), and finally estimating $f(Y \mid \mathbf{X}, \mathbf{Z}, T)$. However, finding a well-defined shadow variable in real-world scenarios is also challenging because it requires domain-specific knowledge of experts and must be investigated on a case-by-case basis (Li et al., 2023). To relax the assumption that prior knowledge about shadow variables is needed, we propose a novel ShadowCatcher to generate representations serving the role of shadow variables directly from observed covariates without prior knowledge and a novel ShadowEstimator to estimate CATE under collider bias with the help of the generated shadow variable representations.

### 3.3 SHADOWCATCHER

Intuitively, as shown in Figure 1(c), the causal link $\mathbf{X} \to \mathbf{Z}$ indicates that the shadow variable is possible to be learned from the fully observed covariates. Therefore, our proposed ShadowCatcher aims to learn representations $\mathbf{Z}$ from $\mathbf{X}$ that satisfy the shadow variable assumptions. To achieve this goal, we must ensure that the generated representations do satisfy Assumption 1.

As stated in Assumption 1, a valid shadow variable needs to satisfy two conditional independence assumptions: (1) $\mathbf{Z} \not\perp\!\!\!\perp Y \mid \mathbf{X}, T, S = 1$, (2) $\mathbf{Z} \perp\!\!\!\perp S \mid \mathbf{X}, T, Y$. The first assumption can be easily tested with only the observed data because only $S = 1$ data is involved. However, the second assumption needs $Y$ to be fully observed, but the fact is that $Y$ values are missing for $S = 0$ data. Fortunately, this assumption is proven refutable with only the observed data.

**Theorem 2 (d'Haultfoeuille, 2010).** Suppose the overlap assumption and $\mathbf{Z} \not\perp\!\!\!\perp Y \mid \mathbf{X}, T, S = 1$ hold, then $\mathbf{Z} \perp\!\!\!\perp S \mid \mathbf{X}, T, Y$ can be rejected if and only if there does not exist any function $Q(\cdot)$ that satisfies the following equation and takes value between $(0, 1]$:

$$\mathbb{E}[S/Q(\mathbf{X}, T, Y) - 1 \mid \mathbf{X}, \mathbf{Z}, T] = 0. \quad (6)$$

Note that Equation (6) only involves the observed data since $\mathbf{X}, \mathbf{Z}, T$ are fully observed and $S/Q(\mathbf{X}, T, Y) = 0$ when $S = 0$. Hence, although we cannot directly test whether the generated $\mathbf{Z}$ satisfies the second assumption, we can test whether the generated $\mathbf{Z}$ can be rejected by Equation (6). As a result, we can tell ShadowCatcher generates valid shadow-variable representations if and only if the generated $\mathbf{Z}$ is tested to be not refutable.

Therefore, ShadowCatcher iteratively generates shadow-variable representations and tests whether the generated representations satisfy Assumption 1 until the generated representations can pass the hypothesis test, detailed as follows.

**Generation Phase.** During the generation process, ShadowCatcher uses a representations generator $g(\mathbf{X}) \to \mathbf{Z}$ to learn representations $\mathbf{Z}$ from $\mathbf{X}$ with the following two constraints:

**(1) Constraining $\mathbf{Z} \not\perp\!\!\!\perp Y \mid \mathbf{X}, T, S = 1$ by a selected outcome estimator.** This estimator aims to estimate $f(Y \mid \mathbf{X}, \mathbf{Z}, T, S = 1)$ with $S = 1$ samples and generated $Z$. The objective is to learn a function $h_{y_1}(\mathbf{X}, \mathbf{Z}, T) \to Y$ by minimize the Mean-Square Error (MSE) between $h_{y_1}(\mathbf{X}_{S=1}, \mathbf{Z}_{S=1}, T_{S=1})$

and $Y_{S=1}$, where $\mathbf{X}_{S=1}$, $\mathbf{Z}_{S=1}$, $T_{S=1}$, and $Y_{S=1}$ denote the value of the corresponding variables of the $S=1$ data. The **loss function of this estimator** is $L_{\mathrm{y1}} = \frac{1}{n_1} \sum_{i:s_i=1} (h_{\mathrm{y1}}(\mathbf{x}_i, \mathbf{z}_i, t_i) - y_i)^2$, where $n_1$ denotes the number of $S=1$ units in $\mathcal{D}$. *Note that ShadowEstimator also uses this estimator.* To constrain the generated $\mathbf{Z}$ satisfying $\mathbf{Z} \not\perp\!\!\!\perp Y \mid \mathbf{X}, T, S=1$, we need to make $f(Y \mid \mathbf{X}, \mathbf{Z}, T, S=1)$ differ from $f(Y \mid \mathbf{X}, \mathbf{Z}^-, T, S=1)$, where $\mathbf{Z}^-$ denotes a value that differs significantly from $\mathbf{Z}$, e.g., for binary $\mathbf{Z}$, $\mathbf{Z}^- = 1 - \mathbf{Z}$; for continuous $\mathbf{Z}$, $\mathbf{Z}^-$ can be a random $\mathbf{Z}$. Therefore, **one objective of the generator** is to simultaneously minimize the MSE between $h_{\mathrm{y1}}(\mathbf{X}_{S=1}, \mathbf{Z}_{S=1}, T_{S=1})$ and $Y_{S=1}$, and maximize the MSE between $h_{\mathrm{y1}}(\mathbf{X}_{S=1}, \mathbf{Z}_{S=1}^-, T_{S=1})$ and $Y_{S=1}$, where $\mathbf{Z}_{S=1}^-$ denotes $\mathbf{Z}^-$ of the $S=1$ data, i.e., to minimize the following loss function:

$$L_{\mathrm{g_y}} = \frac{1}{n_1} \cdot \sum_{i:s_i=1} (h_{\mathrm{y1}}(\mathbf{x}_i, \mathbf{z}_i, t_i) - y_i)^2 - \frac{1}{n_1} \sum_{i:s_i=1} (h_{\mathrm{y1}}(\mathbf{x}_i, \mathbf{z}_i^-, t_i) - y_i)^2.$$

**(2) Constraining $\mathbf{Z} \perp\!\!\!\perp S \mid \mathbf{X}, T, Y$ by a representations estimator.** This estimator aims to estimate $f(\mathbf{Z} \mid \mathbf{X}, T, Y, S=1)$ with $S=1$ samples and generated $Z$. The objective is to learn a function $h_{\mathrm{r}}(\mathbf{X}, T, Y) \to \mathbf{Z}$ by minimize the MSE between $h_{\mathrm{r}}(\mathbf{X}_{S=1}, T_{S=1}, Y_{S=1})$ and $\mathbf{Z}_{S=1}$. The **loss function of this estimator** is $L_{\mathrm{r}} = \frac{1}{n_1} \sum_{i:s_i=1} (h_{\mathrm{r}}(\mathbf{x}_i, t_i, y_i) - \mathbf{z}_i)^2$. To constrain the generated $\mathbf{Z}$ satisfying the $\mathbf{Z} \perp\!\!\!\perp S \mid \mathbf{X}, T, Y$, we need to make $f(\mathbf{Z} \mid \mathbf{X}, T, Y, S=1)$ the same as $f(\mathbf{Z} \mid \mathbf{X}, T, Y, S=0)$. Therefore, **the other objective of the generator** is to minimize the MSE between $h_{\mathrm{r}}(\mathbf{X}_{S=0}, T_{S=0}, Y_{S=0})$ and $\mathbf{Z}_{S=0}$, where $\mathbf{X}_{S=0}$, $\mathbf{Z}_{S=0}$, $T_{S=0}$, and $Y_{S=0}$ denote the value of the corresponding variables of the $S=0$ data., i.e., to minimize the following loss function:

$$L_{\mathrm{g_z}} = \frac{1}{n_0} \cdot \sum_{i:s_i=0} (h_{\mathrm{r}}(\mathbf{x}_i, t_i, h_{\mathrm{y1}}(\mathbf{x}_i, \mathbf{z}_i, t_i)) - \mathbf{z}_i)^2,$$

where $n_0$ denotes the number of $S=0$ units in $\mathcal{D}$. Since the $Y$ values are missing for $S=0$ units, here we use $\hat{Y}_{S=0}$ predicted by $h_{\mathrm{y1}}$ as substitutes. This imputation approach may harm the constraining process, but we can control this impact in the subsequent hypothesis test phase.

Therefore, **the total loss of the representations generator** is $L_{\mathrm{g}} = L_{\mathrm{g_y}} + L_{\mathrm{g_z}}$.

**Hypothesis Test Phase.** In the generation process, the $\mathbf{Z} \perp\!\!\!\perp S \mid \mathbf{X}, T, Y$ assumption is not strictly constrained due to the missing $Y$ values for $S=0$ units. Therefore, ShadowCatcher conducts an additional hypothesis test based on Theorem 2 after the generation phase finishes. The tester aims to learn a solution $q$ of $Q(\mathbf{X}, T, Y)$ in Equation (6) that belongs to $(0, 1]$ which turns into an optimization problem by minimizing

$$L_{\mathrm{q}} = \frac{1}{n} \cdot \sum_{i=1}^{n} \| (s_i / q(\mathbf{x}_i, t_i, y_i) - 1) \cdot (\mathbf{x}_i, \mathbf{z}_i, t_i) \|_2^2,$$

where $q(\mathbf{x}_i, t_i, y_i)$ is a function from $\mathbb{R}$ to $(0, 1]$ and $\| \cdot \|_2$ denotes the $\ell_2$ norm. Note that for $s_i = 0$ units, the value of $s_i / q(\mathbf{x}_i, t_i, y_i)$ equals 0, and thus, the entire optimization process does not involve missing $y_i$ values. Therefore, when the loss function converges, if the loss value is greater than a given threshold $\alpha$, which means it fails to learn a $q$ that satisfies Equation (6), we can tell that no solution of Equation (6) belongs to $(0, 1]$ and Assumption 1 is rejected. Note that to preempt the possible multiple comparisons issue, we use Bonferroni correction (Dunn, 1961) to dynamically adjust $\alpha$ during training by setting $\alpha$ to $\frac{\alpha}{m}$ in the $m$-th iteration. As a result, the generated $\mathbf{Z}$ does not satisfy Assumption 1, and we need to regenerate it until it can pass the hypothesis test, i.e., the converged loss value is less than $\alpha$. Finally, the first generated $\mathbf{Z}$ that passes the test can serve the role of shadow variables and be used for treatment effect estimation under collider bias by ShadowEstimator.

### 3.4 ShadowEstimator

With the help of the generated shadow-variable representations, we can estimate treatment effects under collider bias through: 1) estimating $\widetilde{\mathrm{OR}}(\mathbf{X}, T, Y)$ and $\mathrm{OR}(\mathbf{X}, T, Y)$ by Equation (4, 5); 2) using Equation (3) to recover and estimate $f(Y \mid \mathbf{X}, \mathbf{Z}, T, S=0)$; 3) estimating $f(S \mid \mathbf{X}, \mathbf{Z}, T)$; 4) estimating $f(Y \mid \mathbf{X}, \mathbf{Z}, T)$ and the CATE using estimated $f(Y \mid \mathbf{X}, \mathbf{Z}, T, S=0)$, $f(Y \mid \mathbf{X}, \mathbf{Z}, T, S=1)$ and $f(S \mid \mathbf{X}, \mathbf{Z}, T)$. Note that $f(Y \mid \mathbf{X}, \mathbf{Z}, T, S=1)$ is available from ShadowCatcher.

**Estimation of $\widetilde{\mathrm{OR}}(\mathbf{X}, T, Y)$ and $\mathrm{OR}(\mathbf{X}, T, Y)$.** With the generated $\mathbf{Z}$ and fully observed $\mathbf{X}$ and $T$, we first use two shadow-variable estimator $h_{\mathrm{z0}}(\mathbf{X}, T)$ and $h_{\mathrm{z1}}(\mathbf{X}, T)$ to estimate $f(\mathbf{Z} \mid \mathbf{X}, T, S=0)$ and $f(\mathbf{Z} \mid \mathbf{X}, T, S=1)$ respectively by minimizing the following loss functions:

$$L_{\mathrm{z0}} = \frac{1}{n_0} \cdot \sum_{i:s_i=0} (h_{\mathrm{z0}}(\mathbf{x}_i, t_i) - \mathbf{z}_i)^2, \quad L_{\mathrm{z1}} = \frac{1}{n_1} \cdot \sum_{i:s_i=1} (h_{\mathrm{z1}}(\mathbf{x}_i, t_i) - \mathbf{z}_i)^2.$$

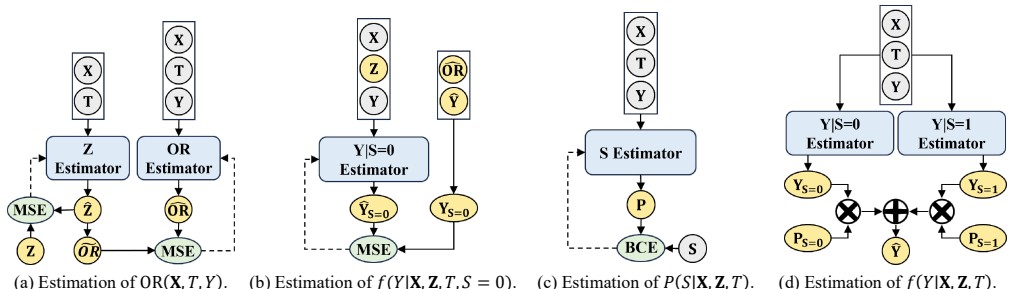

Figure 3: The flowchart of ShadowEstimator, including four estimation procedures.

Using $\mathbf{X}$, $T$, and $Y$ of the $S = 1$ units and $h_{z_0}(\mathbf{X}, T)/h_{z_1}(\mathbf{X}, T)$ as the ground truths, we then estimate $\widetilde{\mathrm{OR}}(\mathbf{X}, T, Y)$ by minimizing the following loss function:

$$L_{\widetilde{\mathrm{or}}} = \frac{1}{n_1} \cdot \sum\nolimits_{i:s_i=1} (\widetilde{\mathrm{or}}(\mathbf{x}_i, t_i, y_i) - h_{z_0}(\mathbf{x}_i, t_i)/h_{z_1}(\mathbf{x}_i, t_i))^2,$$

where $\widetilde{\mathrm{or}}(\cdot)$ is the estimated $\widetilde{\mathrm{OR}}(\cdot)$. Then we can obtain $\mathrm{OR}(\mathbf{X}, T, Y)$ with $\widetilde{\mathrm{or}}(\cdot)$ by Equation (5).

**Estimation of $f(Y \mid \mathbf{X}, \mathbf{Z}, T, S = 0)$.** With the estimated $\mathrm{OR}(\mathbf{X}, T, Y)$, $f(Y \mid \mathbf{X}, \mathbf{Z}, T, S = 1)$ and $\mathbb{E}[\widetilde{\mathrm{OR}}(\mathbf{X}, T, Y) \mid \mathbf{X}, \mathbf{Z}, T, S = 1]$ equaling $\frac{h_{z_0}(\mathbf{X}, T)}{h_{z_1}(\mathbf{X}, T)}$, the ground truth $f(Y \mid \mathbf{X}, \mathbf{Z}, T, S = 0)$ of $S = 1$ samples can be obtained by Equation (3). Therefore, we can learn a function $h_{y_0}(\mathbf{X}, \mathbf{Z}, T) \rightarrow Y$ to estimate $f(Y \mid \mathbf{X}, \mathbf{Z}, T, S = 0)$ using $S = 1$ samples by minimizing the following loss function:

$$L_{y_0} = \frac{1}{n_1} \cdot \sum\nolimits_{i:s_i=1} \left( h_{y_0}(\mathbf{x}_i, \mathbf{z}_i, t_i) - \frac{\widetilde{\mathrm{or}}(\mathbf{x}_i, t_i, y_i) \cdot h_{y_1}(\mathbf{x}_i, \mathbf{z}_i, t_i) \cdot h_{z_1}(\mathbf{x}_i, t_i)}{\widetilde{\mathrm{or}}(\mathbf{x}_i, t_i, 0) \cdot h_{z_0}(\mathbf{x}_i, t_i)} \right)^2.$$

**Estimation of $f(Y \mid \mathbf{X}, \mathbf{Z}, T)$.** Now that $f(Y \mid \mathbf{X}, \mathbf{Z}, T, S = 0)$ and $f(Y \mid \mathbf{X}, \mathbf{Z}, T, S = 1)$ are both estimated, estimation of $f(Y \mid \mathbf{X}, \mathbf{Z}, T)$ becomes estimation of $f(S \mid \mathbf{X}, \mathbf{Z}, T)$, which can be achieved by minimizing the following loss function using fully observed $\mathbf{X}$, $\mathbf{Z}$ and $T$ to learn a function $h_s(\mathbf{X}, \mathbf{Z}, T) \rightarrow S$:

$$L_s = -\frac{1}{n} \cdot \sum_{i=1}^{n} (s_i \cdot \log(h_s(\mathbf{x}_i, \mathbf{z}_i, t_i)) + (1 - s_i) \cdot \log(1 - h_s(\mathbf{x}_i, \mathbf{z}_i, t_i))),$$

and then we can obtain $f(Y \mid \mathbf{X}, \mathbf{Z}, T)$ by:

$$f(Y \mid \mathbf{X}, \mathbf{Z}, T) = \sum\nolimits_{s \in \{0,1\}} f(Y \mid \mathbf{X}, \mathbf{Z}, T, S = s) \cdot f(S = s \mid \mathbf{X}, \mathbf{Z}, T). \tag{7}$$

Then, we can use Equation (1) to achieve CATE estimation under collider bias. Note that we apply existing de-confounding methods (Shalit et al., 2017) to the outcome estimators during training to address possible confounding bias. The pseudo-codes and the overall flowchart are in Appendix A.1.

## 4 EXPERIMENTS

### 4.1 BASELINES

As stated in Section 2, there is currently no causal inference method that can solve collider bias without introducing additional assumptions and prior knowledge. Therefore, we implement the following treatment effect estimators that focus on confounding bias and sample selection bias caused by $\mathbf{X}$ and $T$ as our baselines: (1) Heckman's Correction (Heckit) (Heckman, 1979), (2) Doubly Robust (Bang & Robins, 2005), (3) Inverse Probability of Sampling Weights (IPSW) (Cole & Stuart, 2010), (4) Balancing Neural Network (BNN) (Johansson et al., 2016), (5) Treatment-Agnostic Representation Network (TARNet), (6) CounterFactual Regression (CFR) (Shalit et al., 2017), (7) Causal Forest (CForest) (Wager & Athey, 2018), (8) Disentangled Representations for CounterFactual Regression (DR-CFR) (Greiner, 2020), (9) TEDVAE (Zhang et al., 2021), (10) Decomposed Representations for CounterFactual Regression (DeR-CFR) (Wu et al., 2022), (11)

Table 1: The results of CATE estimation ($\sqrt{\text{PEHE}}$) on synthetic datasets under different $\beta$.

| Estimator | $\beta = 1$ | | $\beta = 3$ | | $\beta = 5$ | |
|---|---|---|---|---|---|---|
| | Selected data | Unselected data | Selected data | Unselected data | Selected data | Unselected data |
| Heckit | 0.323±0.065 | 0.330±0.046 | 0.340±0.055 | 0.352±0.042 | 0.349±0.069 | 0.413±0.048 |
| DR | 0.298±0.032 | 0.316±0.042 | 0.331±0.048 | 0.357±0.053 | 0.367±0.033 | 0.448±0.017 |
| IPSW | 0.328±0.048 | 0.348±0.049 | 0.328±0.031 | 0.353±0.034 | 0.465±0.011 | 0.545±0.014 |
| BNN | 0.290±0.011 | 0.306±0.012 | 0.329±0.048 | 0.354±0.033 | 0.359±0.011 | 0.439±0.015 |
| TARNet | 0.295±0.012 | 0.312±0.011 | 0.329±0.030 | 0.357±0.053 | 0.366±0.071 | 0.436±0.087 |
| CFR | 0.290±0.009 | 0.307±0.008 | 0.324±0.009 | 0.350±0.013 | 0.359±0.008 | 0.436±0.030 |
| CForest | 0.310±0.030 | 0.331±0.038 | 0.338±0.019 | 0.368±0.022 | 0.373±0.026 | 0.453±0.043 |
| DR-CFR | 0.284±0.038 | 0.307±0.040 | 0.340±0.055 | 0.355±0.064 | 0.366±0.051 | 0.435±0.060 |
| TEDVAE | 0.281±0.056 | 0.419±0.070 | 0.378±0.063 | 0.420±0.059 | 0.394±0.054 | 0.431±0.067 |
| DeR-CFR | 0.291±0.010 | 0.309±0.014 | 0.323±0.015 | 0.348±0.017 | 0.358±0.011 | 0.439±0.013 |
| DESCN | 0.295±0.002 | 0.312±0.002 | 0.326±0.003 | 0.357±0.004 | 0.365±0.003 | 0.449±0.011 |
| ES-CFR | 0.289±0.003 | 0.305±0.004 | 0.331±0.003 | 0.359±0.003 | 0.369±0.003 | 0.448±0.005 |
| Ours | 0.241±0.014 | 0.248±0.009 | 0.305±0.013 | 0.326±0.015 | 0.333±0.040 | 0.404±0.053 |
| Ours (New) | **0.227±0.001** | **0.229±0.001** | **0.249±0.013** | **0.255±0.021** | **0.299±0.008** | **0.300±0.008** |

Deep Entire Space Cross Networks (DESCN) (Zhong et al., 2022), (12) Entire Space CounterFactual Regression (ES-CFR) (Wang et al., 2023) to estimate the CATE and compare them with our proposed methods. Based on the estimated CATE, we use the Precision in Estimation of Heterogeneous Effect (PEHE) (Shalit et al., 2017; Louizos et al., 2017) to evaluate the performance of the above methods, where $\text{PEHE} = \frac{1}{N} \cdot \sum_{i=1}^{N}((\hat{y}_i(1) - \hat{y}_i(0)) - (y_i(1) - y_i(0))^2$. We split each dataset into 60/20/20 train/validation/test datasets, independently repeat 20 times, and report the mean and standard deviation (std) of $\sqrt{\text{PEHE}}$ for all experiments, formed as mean ± std in the tables.

## 4.2 EXPERIMENTS ON SYNTHETIC DATA

### 4.2.1 DATASETS

In order to better evaluate the performance of each estimator under collider bias, we generate synthetic datasets with different collider bias strengths, denoted by $\beta$, which affects the impact of $Y$ on $S$. The size $n$ of all datasets is 10,000, and the dimension $d$ of the covariates is 10. To compare our methods with the baselines under different strengths of collider bias, we set $d_{\text{s}} = 0.9 \cdot d$ and evaluate the performance of each estimator under $\beta = \{1, 3, 5\}$. We also conduct additional experiments on the synthetic data for evaluating the impact of different non-shadow-variables proportions in the covariates, the impact of the reject threshold $\alpha$ of ShadowCatcher, and the effectiveness of the constraints in the generation phase of ShadowCatcher. The data generation process and the additional experiments are detailed in Appendix A.2.

### 4.2.2 RESULTS

We separately report the results of the selected data ($S = 1$) and unselected data ($S = 0$) in Table 1 under different collider bias strengths with $\beta = \{1, 3, 5\}$. We observe that: (1) The overall performance of DR, BNN, CFR, CForest, TEDVAE, DR-CFR, DESCN, DeR-CFR and ES-CFR is not good because they all focus on confounding bias and thus cannot deal with sample selection bias. (2) The performance of Heckit and IPSW is also poor because they can only address sample selection bias caused by $T$ and $\mathbf{X}$ and cannot address collider bias because of the spurious association $T \to S \leftarrow Y$. (3) Our method outperforms all baselines under all $\beta$ settings because the generated representations by ShadowCatcher make identification under collider bias possible, and ShadowEstimator provides a practical solution. (4) As collider bias strengthens, the performance gap between selected and unselected data increases because the more substantial the collider bias is, the more significant the distribution shift problem is. However, this gap for our method is much smaller than that of other baselines, which demonstrates that our proposed approaches can practically address collider bias.

## 4.3 EXPERIMENTS ON REAL-WORLD DATA

### 4.3.1 DATASETS

In order to evaluate the proposed method in real-world scenarios, we conduct experiments on three well-known datasets: **the IHDP dataset** (Hill, 2011), **the ACIC 2016 dataset** (Dorie et al.,

Table 2: The results of CATE estimation on three real-world datasets.

| Estimator | IHDP ($\sqrt{\text{PEHE}}$) | | ACIC 2016 ($\sqrt{\text{PEHE}}$) | | Jobs ($\hat{R}_{\text{Pol}}$) | |
|---|---|---|---|---|---|---|
| | Within-sample | Out-of-sample | Within-sample | Out-of-sample | Within-sample | Out-of-sample |
| Heckit | 1.587±0.065 | 1.621±0.041 | 3.106±0.444 | 3.340±0.111 | 0.328±0.050 | 0.331±0.052 |
| DR | 1.355±0.123 | 1.572±0.205 | 2.346±0.129 | 2.653±0.222 | 0.316±0.007 | 0.317±0.036 |
| IPSW | 2.118±0.344 | 2.129±0.295 | 4.244±0.145 | 5.411±0.073 | 0.284±0.051 | 0.289±0.063 |
| BNN | 1.308±0.298 | 1.457±0.339 | 2.173±0.150 | 2.586±0.486 | 0.303±0.025 | 0.304±0.041 |
| TARNet | 1.240±0.158 | 1.416±0.154 | 2.275±0.756 | 2.805±0.766 | 0.315±0.012 | 0.316±0.050 |
| CFR | 1.283±0.186 | 1.401±0.238 | 2.107±0.297 | 2.361±0.587 | 0.313±0.018 | 0.314±0.072 |
| CForest | 1.702±0.292 | 1.948±0.429 | 4.137±0.295 | 4.605±0.137 | 0.326±0.012 | 0.326±0.059 |
| DR-CFR | 1.299±0.087 | 1.399±0.171 | 2.240±0.691 | 2.340±0.663 | 0.322±0.022 | 0.323±0.099 |
| TEDVAE | 4.246±0.394 | 4.347±0.563 | 3.501±0.708 | 4.468±0.813 | 0.296±0.046 | 0.300±0.031 |
| DeR-CFR | 1.446±0.345 | 1.571±0.371 | 2.214±0.204 | 2.246±0.598 | 0.309±0.023 | 0.311±0.029 |
| DESCN | 1.193±0.057 | 1.665±0.246 | 2.185±0.150 | 2.306±0.236 | 0.331±0.010 | 0.331±0.051 |
| ES-CFR | 1.499±0.096 | 1.436±0.095 | 3.875±0.224 | 4.494±0.214 | 0.290±0.045 | 0.293±0.046 |
| Ours | 1.039±0.069 | 1.065±0.099 | 2.078±0.333 | 2.142±0.390 | 0.283±0.018 | 0.284±0.080 |
| Ours (New) | **0.703±0.106** | **0.723±0.102** | **1.911±0.126** | **2.047±0.351** | **0.279±0.017** | **0.280±0.018** |

2019), and **the Jobs dataset** (Shalit et al., 2017)[5]. The ground truth CATE is known in the IHDP and ACIC 2016 datasets, so we use the same metric as those in the experiments on the synthetic data. Following Shalit et al. (2017), since the ground truth CATE is unknown in the Jobs dataset, we use the policy risk to evaluate the quality of CATE estimation. The policy risk is defined as the average loss in value when treating according to the policy implied by a CATE estimator: $\hat{R}_{\text{Pol}} = 1 - (\mathbb{E}[Y(1) \mid \tau(\mathbf{x}) > 0, T = 1] \cdot \mathbb{P}(\tau(\mathbf{x}) > 0) + \mathbb{E}[Y(0) \mid \tau(\mathbf{x}) \leq 0, T = 0] \cdot \mathbb{P}(\tau(\mathbf{x}) \leq 0))$. We report the mean and std of the policy risk formed as mean $\pm$ std in the table. More details about these datasets and the simulation process are provided in Appendix A.2.3.

### 4.3.2 RESULTS

We separately report the results of within-sample data and out-of-sample data in Table 2, where within-sample means that the (factual) outcome of one treatment is observed, i.e., the $S = 1$ samples for training, and out-of-sample means no observed outcomes, i.e., the $S = 1$ samples for testing and all $S = 0$ samples (Shalit et al., 2017). From the results, we observe that: (1) The performance of the methods on confounding bias is not good because they cannot address sample selection bias. (2) The performance of the methods on sample selection bias is also poor because they can only address the cases that $\mathbf{X}$ and $T$ cause $S$ and thus cannot achieve a better estimate under collider bias. (3) Our method outperforms all baselines on both datasets because ShadowCatcher and ShadowEstimator effectively address collider bias in data. (4) The performance gap between our proposed method's within-sample and out-of-sample data is also overall the lowest, proving the ability to counterfactual prediction of our method. (5) The proposed method on the Jobs dataset shows the lowest policy risk, which demonstrates the effectiveness of our methods in real-world applications.

## 5 CONCLUSION

In this paper, we overcome the challenge of finding valid shadow variables to estimate treatment effects under collider bias in observational studies. We propose a novel ShadowCatcher that can generate representations serving the role of shadow variables and a novel ShadowEstimator that uses the generated representations to estimate CATE under collider bias. Experimental results demonstrate the effectiveness and application value of ShadowCatcher and ShadowEstimator. One main limitation of our work is that the choice of the reject threshold $\alpha$ is a tradeoff between efficiency and performance during the generation process of ShadowCatcher. The impact of different options of $\alpha$ on the efficiency and performance of ShadowCatcher is further discussed in Appendix A.2.

---

[5]The IHDP dataset is available at http://www.fredjo.com/; The ACIC 2016 dataset is available at https://github.com/vdorie/aciccomp/tree/master/2016; The Jobs dataset is available at https://users.nber.org/~rdehejia/nswdata2.html.

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

# A APPENDIX

## A.1 PSEUDO-CODES OF SHADOWCATCHER AND SHADOWESTIMATOR

As stated in Section 3, we propose a novel ShadowCatcher that generates representations serving the role of shadow variables and a novel ShadowEstimator that estimates treatment effects under collider bias with the help of the generated representations. The pseudo-codes of ShadowCatcher and ShadowEstimator are detailed in Algorithm 1 and 2, where $g$ denotes the representations generator, $h_{y_1}$ denotes the selected outcome estimator, $h_{y_0}$ denotes the unselected outcome estimator, $h_r$ denotes the representations estimator, $h_{z_1}$ and $h_{z_0}$ denote the shadow-variable estimators, $\widetilde{or}$ denotes the odds ratio estimator, $h_s$ denotes the sample selection estimator, and $q$ denotes the $Q$ function solver.

---

**Algorithm 1:** ShadowCatcher

---
**Data:** the observational dataset $\mathcal{D}$, reject threshold $\alpha$.
**Result:** the observational dataset $\mathcal{D}$ with the generated $\mathbf{Z}$.
$l_q \leftarrow \alpha + 1$;
$m \leftarrow 1$;
initialization of parameters in $h_{y_1}$, $h_r$, $q$ and $g$;
**while** $l_q \geq \alpha$ **do**
   $\alpha \leftarrow \alpha/m$;
   $m \leftarrow m + 1$;
   Generation Phase: use mini-batch gradient descent to simutaneously

      ❶ optimize $h_{y_1}$ by minimizing $L_{y_1}$ with $S = 1$ units in $\mathcal{D}$ and the generated $\mathbf{Z}$ by $g$;
      ❷ estimate the missing $Y$ values for $S = 0$ units as replacements of the missing values in $\mathcal{D}$;
      ❸ optimize $h_r$ by minimizing $L_r$ with $S = 1$ units in $\mathcal{D}$ and the generated $\mathbf{Z}$ by $g$;
      ❹ optimize $g$ by minimizing $L_g = L_{g_y} + L_{g_z}$ with all units in $\mathcal{D}$ and the generated $\mathbf{Z}$ by $g$;

   Hypothesis Test Phase: use mini-batch gradient descent to optimize $q$ by minimizing $L_q$;
   insert the generated $\mathbf{Z}$ by $g$ into $\mathcal{D}$;
   update $l_q$ with the final output of $L_q$;
**end**

---

---

**Algorithm 2:** ShadowEstimator

---
**Data:** the observational dataset $\mathcal{D}$ with the generated $\mathbf{Z}$.
**Result:** the CATE of all units in $\mathcal{D}$.
initialization of parameters in $h_{y_0}$, $h_{z_0}$, $h_{z_1}$, $\widetilde{or}$ and $h_s$;
use mini-batch gradient descent to sequentially

❶ optimize $h_{z_0}$ and $h_{z_1}$ by minimizing $L_{z_0}$ and $L_{z_1}$;

❷ calculate the "ground truth" values of $\widetilde{\mathrm{OR}}(\mathbf{X}, T, Y)$ by Equation (4);

❸ optimize $\widetilde{or}$ by minimizing $L_{or}$ with $S = 1$ units in $\mathcal{D}$ and the calculated "ground truth" values;

❹ calculate the "ground truth" values of $\mathrm{OR}(\mathbf{X}, T, Y)$ by Equation (5);

❺ calculate the "ground truth" values of $Y$ values of $S = 0$ units in $\mathcal{D}$ by Equation (3);

❻ optimize $h_{y_0}$ by minimizing $L_{y_0}$ with $S = 0$ units in $\mathcal{D}$ and the calculated "ground truth" values;

❼ optimize $h_s$ by minimizing $L_s$ with all units in $\mathcal{D}$;

calculate the CATE of all units in $\mathcal{D}$ with the optimized $h_{y_0}$, $h_{y_1}$ and $h_s$;

---

The overall framework of ShadowCatcher and ShadowEstimator is: ShadowCatcher first takes the fully observed $\mathbf{X}$ and $T$, and the observed $Y$ of $S = 1$ units as inputs to generate shadow-variable representations $\mathbf{Z}$, and then tests whether the generated $\mathbf{Z}$ satisfy Assumption 1. If the generated $\mathbf{Z}$ does not pass the hypothesis test, ShadowCatcher should re-generate new shadow-variable representations until the generated $\mathbf{Z}$ finally passes the test. After that, ShadowEstimator

Table 3: The results of CATE estimation ($\sqrt{\text{PEHE}}$) on the Twins datasets.

| Estimators | Within-sample | Out-of-sample |
|---|---|---|
| Heckit | 0.345±0.023 | 0.357±0.023 |
| DR | 0.476±0.010 | 0.487±0.007 |
| IPSW | 0.339±0.009 | 0.344±0.021 |
| BNN | 0.358±0.021 | 0.373±0.021 |
| TARNet | 0.401±0.049 | 0.407±0.058 |
| CFR | 0.361±0.040 | 0.369±0.040 |
| CForest | 0.356±0.034 | 0.421±0.035 |
| DR-CFR | 0.340±0.028 | 0.350±0.028 |
| TEDVAE | 0.319±0.003 | 0.337±0.008 |
| DeR-CFR | 0.316±0.009 | 0.321±0.013 |
| DESCN | 0.401±0.021 | 0.432±0.029 |
| ES-CFR | 0.312±0.010 | 0.320±0.023 |
| Ours | 0.308±0.003 | 0.311±0.004 |
| Ours (New) | **0.294±0.008** | **0.304±0.015** |

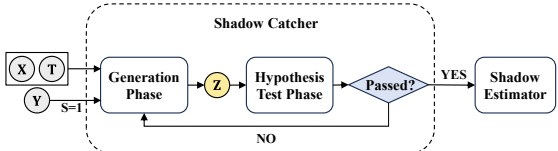

Figure 4: The overall flowchart of ShadowCatcher and ShadowEstimator.

Table 4: The hyperparameters of ShadowCatcher and ShadowEstimator on different datasets.

| Dataset | epochs | batch size | learning rate | weight decay | IPM weight | $\alpha$ |
|---|---|---|---|---|---|---|
| Synthetic datasets | 100 | 1024 | 0.03 | 0.01 | 0.001 | 1e-6 |
| The IHDP dataset | 100 | 128 | 0.03 | 0.01 | 0.001 | 0.01 |
| The Twins dataset | 100 | 1024 | 0.03 | 0.01 | 0.1 | 0.1 |
| The Jobs dataset | 100 | 256 | 0.003 | 0.001 | 0.1 | 0.1 |
| The ACIC 2016 datasets | 100 | 256 | 0.01 | 0.001 | 0.001 | 100 |

uses the generated $\mathbf{Z}$ to estimate treatment effects with observational samples. The overall flowchart is shown in Figure 4.

## A.2 SUPPLEMENT TO THE EXPERIMENTS SECTION

### A.2.1 IMPLEMENTATION DETAILS

We utilize 3-layer Neural Networks to implement each module in ShadowEstimator and ShadowCatcher. We use the Adam optimizer (Kingma & Ba, 2015) with batch normalization (Ioffe & Szegedy, 2015) in the training process, and we use the Wasserstein distance (Cuturi & Doucet, 2014) as the Integral Probability Metric (IPM) to implement all the methods that need IPM to balance representations. The hyperparameters of our methods on different datasets are detailed in Table 4. We implement all the methods in the PyTorch environment with Python 3.9. The CPU is 13th Gen Intel(R) Core(TM) i7-13700K, and the GPU is NVIDIA GeForce RTX 3080 with CUDA 12.1.

### A.2.2 DATA GENERATION PROCESS OF THE SYNTHETIC DATASETS

We first generate the continuous covariates $\mathbf{X} \in \mathbb{R}^{n \times d}$ with independent Gaussian distributions as $\mathbf{X} \stackrel{\text{i.i.d.}}{\sim} N(\mathbf{0}, \mathbf{1})$, and then generate the binary treatment variable $T \in \mathbb{R}^{n \times 1}$ from a logistic function as $T \sim \text{Bernoulli}(1/(1 + e^{-t(\mathbf{X})}))$, where $\text{Bernoulli}(\cdot)$ denotes the Bernoulli distribution, $t(\mathbf{X}) = \sum_{i=1}^{d}(\mathbf{1}(\text{mod}(i,2) \neq 1) - \mathbf{1}(\text{mod}(i,2) \equiv 1)) \cdot X_i/d) + \epsilon_t$, $\mathbf{1}(\cdot)$ is the indicator function, function $\text{mod}(a,b)$ returns the modulus after division of $a$ by $b$ and $\epsilon_t \sim \mathcal{N}(0,1)$. Next, we generate the continuous outcome variable $Y \in \mathbb{R}^{n \times 1}$ from a non-linear function as $Y = \text{Sigmoid}(T + \sum_{i=1}^{d}(T \cdot X_i + (\mathbf{1}(\text{mod}(i,2) \neq 1) - \mathbf{1}(\text{mod}(i,2) \equiv 1)) \cdot (X_i + X_i^2)/d) + \epsilon_y)$, where Sigmoid denotes the sigmoid function and $\epsilon_y \sim \mathcal{N}(0,1)$. To introduce collider bias with strength $\beta$ and implicit shadow variables into datasets, we generate the binary selection variable $S \in \mathbb{R}^{n \times 1}$ from a logistic function $S \sim \text{Bernoulli}(1/(1 + e^{-s(\mathbf{X},T)}))$, where $s(\mathbf{X},T) = T - \beta \cdot Y + \sum_{i=1}^{d_s}(\mathbf{1}(\text{mod}(i,2) \equiv 1) - \mathbf{1}(\text{mod}(i,2) \neq 1)) \cdot X_i/d) + \epsilon_s$ with $\epsilon_s \sim \mathcal{N}(0,1)$. Note that $d_s \leq d$ denotes the dimension of $\mathbf{X}$ that contributes to $S$, and the remaining covariates not involved in the sample selection are implicit shadow variables. A unit is selected into the sample, i.e., the outcome values can be observed only when $S = 1$. The ground truth CATE can be calculated easily by the above functions.

### A.2.3 REAL-WORLD DATASETS DETAILS

The IHDP dataset is from a study evaluating the effect of specialist home visits on the future cognitive test scores of premature infants (Brooksgunn et al., 1992), where confounding bias is introduced by removing a non-random subset of the treated group and using simulated outcomes to replace the original ones. To further introduce collider bias into the IHDP dataset, we set $S = 0$ for $T = 0$ units that the mother boozes and the infant's score is lower than the mean value. Intuitively, unlike the treated group, which can carefully design and regularly follow up to ensure the collection of effective test results, the control group is more likely to have sample selection bias. For those mothers with boozing problems and mothers whose children have weaker cognitive abilities, it is more likely that they will not take their children to participate in the cognitive test, resulting in collider bias. The final IHDP dataset comprises 748 units (557 selected, 191 unselected) with 25 covariates. The ground truth CATE is known because the outcomes are simulated, and both the factual and counterfactual outcomes are available.

The 2016 Atlantic Causal Inference Challenge (ACIC 2016) (Dorie et al., 2019) contains various settings of benchmark datasets with confounding bias simulated by comprehensive data generation processes. To introduce collider bias into the ACIC 2016 datasets, we use the same simulation of $S$ as stated in Section A.2.2: $S \sim \text{Bernoulli}(1/(1 + e^{-s(\mathbf{X},T)}))$, where $s(\mathbf{X}, T) = T - Y + \sum_{i=1}^{d_s}(\mathbf{1}(\text{mod}(i, 2) \equiv 1) - \mathbf{1}(\text{mod}(i, 2) \neq 1)) \cdot X_i/d) + \epsilon_s$ with $\epsilon_s \sim \mathcal{N}(0, 1)$ and $d = 58$.

The Jobs dataset combines a randomized study based on the National Supported Work (NSW) program with observational data to form a larger confounding biased dataset that focuses on estimating the effects of a job training program on future employment situation (LaLonde, 1986; Shalit et al., 2017). To introduce collider bias into the Jobs dataset, we set $S = 0$ for $T = 0$ units that used to have a job but become unemployed. Intuitively, for ones who used to have a job and have not participated in job training programs, it is more likely that they are unwilling to report their current employment situation if they lose their job, leading to collider bias. The final Jobs dataset comprises 2675 units (2494 selected, 181 unselected) with 10 covariates.

The Twins data is from a study evaluating the effect of low birth weight on the mortality of infants in their first year of life (Almond et al., 2005), where confounding bias is introduced by using simulated treatments to replace the original ones (Louizos et al., 2017; Yoon et al., 2018). To introduce collider bias into the Twins dataset, we set $S = 0$ for $T = 1$ units that both the mother uses tobacco and the twin is alive. Intuitively, parents seldom take relatively healthy infants to the hospital, so it is more difficult to record the data of these infants, resulting in collider bias. The final Twins dataset comprises 9643 units (8804 selected, 839 unselected) with 48 covariates. The ground truth CATE is known because, for each twin pair, we observed both the case $T = 0$ (lighter twin) and $T = 1$ (heavier twin) (Yoon et al., 2018). The results are reported in Table 3.

### A.2.4 ADDITIONAL EXPERIMENTAL STUDIES

In addition to the results stated in Section 4.2, we also conduct more experiments detailed as follows:

**Studies of the impact of different non-shadow variables proportions in the covariates.** In Section 4.2, we generate synthetic datasets to evaluate the performance of our proposed ShadowCatcher and ShadowEstimator under different strengths of collider bias, i.e., $\beta$ that affects the impact of $Y$ on $S$. To ensure that the strength of collider bias is only determined by $\beta$, we fix the proportion of non-shadow variables in covariates by setting $d_s = 0.9 \cdot d$. Intuitively, this proportion can also determine the strength of collider bias because it affects how many covariates are involved in the sample selection. The smaller $d_s$ is, the weaker the collider bias is. Therefore, we also conduct experiments under different $d_s$ settings with a fixed $\beta = 5$. The results are in Table 5.

Our observations and analyses are as follows: (1) In general, the performance of all estimators gradually decreases as the proportion of non-shadow variables in covariates increases because the impact of $\mathbf{X}$ on $S$ increases. (2) The performance of IPSW under $d_s = 0.1 \times d$ is abnormally poor because IPSW estimates $\mathbb{P}(S \mid \mathbf{X}, T)$ instead of the ideal $\mathbb{P}(S \mid \mathbf{X}, T, Y)$ for reweighting, the difference of which is significant when the impact of $Y$ on $S$ far exceeds that of $\mathbf{X}$ and $T$ on $S$, leading to an inaccurate estimate. (3) The overall performance of all estimators on selected data is better than unselected data because collider bias results in $\mathbb{E}[Y \mid \mathbf{X}, T, S = 1] \neq \mathbb{E}[Y \mid \mathbf{X}, T, S = 0]$. (4) As the proportion of non-shadow variables in covariates increases, the performance gap between

Table 5: The results of CATE estimation ($\sqrt{\mathrm{PEHE}}$) on synthetic datasets under different $d_{\mathrm{s}}$.

| | $d_{\mathrm{s}} = 0.1 \cdot d$ | | $d_{\mathrm{s}} = 0.5 \cdot d$ | | $d_{\mathrm{s}} = 0.9 \cdot d$ | |
|---|---|---|---|---|---|---|
| Estimator | Selected data | Unselected data | Selected data | Unselected data | Selected data | Unselected data |
| Heckit | 0.100±0.013 | 0.120±0.016 | 0.359±0.044 | 0.367±0.092 | 0.349±0.069 | 0.413±0.048 |
| DR | 0.129±0.022 | 0.130±0.030 | 0.315±0.038 | 0.368±0.058 | 0.367±0.033 | 0.448±0.017 |
| IPSW | 0.604±0.284 | 0.627±0.287 | 0.331±0.060 | 0.353±0.064 | 0.465±0.011 | 0.545±0.014 |
| BNN | 0.103±0.014 | 0.110±0.016 | 0.305±0.007 | 0.358±0.009 | 0.359±0.011 | 0.439±0.015 |
| TARNet | 0.105±0.015 | 0.106±0.021 | 0.307±0.056 | 0.360±0.056 | 0.366±0.071 | 0.436±0.087 |
| CFR | 0.104±0.005 | 0.105±0.017 | 0.307±0.041 | 0.358±0.055 | 0.359±0.008 | 0.436±0.030 |
| CForest | 0.105±0.011 | 0.109±0.012 | 0.312±0.022 | 0.363±0.026 | 0.373±0.026 | 0.453±0.043 |
| DR-CFR | 0.106±0.005 | 0.113±0.011 | 0.287±0.045 | 0.361±0.057 | 0.366±0.051 | 0.435±0.060 |
| TEDVAE | 0.227±0.018 | 0.257±0.021 | 0.283±0.052 | 0.378±0.059 | 0.394±0.054 | 0.431±0.067 |
| DeR-CFR | 0.095±0.011 | 0.097±0.011 | 0.319±0.050 | 0.348±0.017 | 0.358±0.011 | 0.439±0.013 |
| DESCN | 0.107±0.002 | 0.109±0.002 | 0.311±0.004 | 0.367±0.004 | 0.365±0.003 | 0.449±0.011 |
| ES-CFR | 0.094±0.004 | 0.098±0.005 | 0.308±0.002 | 0.360±0.004 | 0.369±0.003 | 0.448±0.005 |
| Ours | 0.088±0.003 | 0.089±0.003 | 0.242±0.010 | 0.279±0.012 | 0.333±0.040 | 0.404±0.053 |
| Ours (New) | **0.085±0.001** | **0.086±0.002** | **0.228±0.006** | **0.256±0.009** | **0.299±0.008** | **0.300±0.008** |

Table 6: The results of CATE estimation ($\sqrt{\mathrm{PEHE}}$) by different versions of ShadowCatcher.

| Version of ShadowCatcher | Selected data | Unselected data |
|---|---|---|
| ShadowCatcher without the constraint on $\mathbf{Z} \not\perp\!\!\!\perp Y \mid \mathbf{X}, T, S = 1$ | 0.288±0.056 | 0.306±0.076 |
| ShadowCatcher with the constraint on $\mathbf{Z} \not\perp\!\!\!\perp Y \mid \mathbf{X}, T, S = 1$ | **0.227±0.001** | **0.229±0.001** |

Table 7: The results of CATE estimation ($\sqrt{\mathrm{PEHE}}$) on synthetic datasets with different options of $\alpha$.

| Reject threshold | Selected data | Unselected data |
|---|---|---|
| $1e-4$ | 0.235±0.003 | 0.235±0.008 |
| $5e-5$ | 0.228±0.003 | 0.231±0.004 |
| $1e-5$ | 0.228±0.002 | 0.231±0.001 |
| $5e-6$ | 0.227±0.002 | 0.230±0.002 |
| $1e-6$ | **0.227±0.001** | **0.229±0.001** |

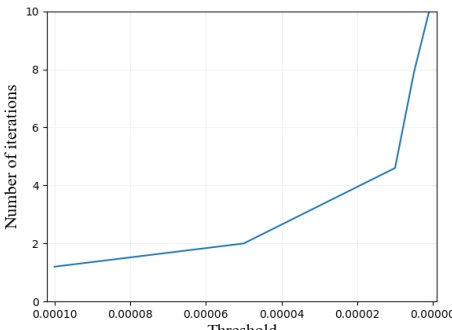

Figure 5: The mean number of iterations for ShadowCatcher to pass the hypothesis test phase with different options of $\alpha$.

selected and unselected data increases because the more substantial the collider bias is, the more significant the distribution shift problem is. Especially when only one covariate is involved in the sample selection, the gap nearly disappears for most estimators. (5) Our method outperforms all baselines under all $d_\text{s}$ settings, and the performance gap between selected data and unselected data, though it still exists, is much smaller than that of other baselines, which demonstrates that our proposed approaches can practically address collider bias in CATE estimation.

**Studies of the effectiveness of the constraints in the generation phase of ShadowCatcher.** During the generation phase of ShadowCatcher, we make two constraints on the representations generator to ensure that the learned representations satisfy the assumptions of shadow variables. The constraint on $\mathbf{Z} \perp\!\!\!\perp S \mid \mathbf{X}, T, Y$ assumption is already guaranteed effective by the hypothesis test phase. However, the effectiveness of the constraint on $\mathbf{Z} \not\perp\!\!\!\perp Y \mid \mathbf{X}, T, S = 1$ assumption still needs to be proved. Therefore, we conduct ablation studies by comparing the performance of ShadowCatcher with and without the constraint on $\mathbf{Z} \not\perp\!\!\!\perp Y \mid \mathbf{X}, T, S = 1$. Specifically, the ablation version of ShadowCatcher optimizes the generator and the selected outcome estimator only by minimizing $L_{\text{g}_\text{z}}$ and $L_{\text{y}_1}$. We conduct the experiments on the synthetic dataset in Section 4.2.1 with $d_\text{s} = 0.9 \cdot d$, $\alpha = 1e-6$, and $\beta = 1$. The results are in Table 6. The results show that the performance of the ablation version of ShadowCatcher gets worse, though still better than other baselines reported in Table 1, proving the effectiveness and necessity of the constraints in the generation phase of ShadowCatcher.

**Studies of the impact of the reject threshold $\alpha$.** As stated in Section 5, the choice of the reject threshold $\alpha$ is a tradeoff between efficiency and performance during the generation process of ShadowCatcher: if the reject threshold is too small, the generated representations may be too weak to be a valid shadow variable; if the threshold is too large, it may needs more iterations for the generated representations to pass the test. To further study the impact of different options of $\alpha$ on the efficiency and performance of ShadowCatcher, we conduct experiments with $\alpha = \{1e-4, 5e-5, 1e-5, 5e-6, 1e-6\}$ on the synthetic dataset in Section 4.2.1 with $d_\text{s} = 0.9 \cdot d$ and $\beta = 1$. The results are in Table 7 and Figure 5. The results show that the performance of ShadowCatcher improves as the reject threshold decreases because the hypothesis test gets more strict, which means the constraint gets more reliable. However, the number of iterations required for ShadowCatcher to pass the hypothesis test also increases very quickly, making the efficiency of ShadowCatcher reduced. Therefore, choosing an appropriate $\alpha$ is a tradeoff between efficiency and performance and depends on the real application scenarios.

### A.3 FURTHER EXPLANATIONS OF SOME FORMULAS

#### A.3.1 AN EXPLANATION OF EQUATION (2)

In Equation (2), the original odds ratio function is

$$
\begin{aligned}
\text{OR}(\mathbf{X}, \mathbf{Z}, T, Y) =& \frac{f(Y \mid \mathbf{X}, \mathbf{Z}, T, S=0) \cdot f(Y=0 \mid \mathbf{X}, \mathbf{Z}, T, S=1)}{f(Y \mid \mathbf{X}, \mathbf{Z}, T, S=1) \cdot f(Y=0 \mid \mathbf{X}, \mathbf{Z}, T, S=0)} \\
=& \frac{f(Y \mid \mathbf{X}, \mathbf{Z}, T, S=0) \cdot f(\mathbf{X}, \mathbf{Z}, T, S=0) \cdot f(Y=0 \mid \mathbf{X}, \mathbf{Z}, T, S=1) \cdot f(\mathbf{X}, \mathbf{Z}, T, S=1)}{f(Y \mid \mathbf{X}, \mathbf{Z}, T, S=1) \cdot f(\mathbf{X}, \mathbf{Z}, T, S=1) \cdot f(Y=0 \mid \mathbf{X}, \mathbf{Z}, T, S=0) \cdot f(\mathbf{X}, \mathbf{Z}, T, S=0)} \\
=& \frac{f(Y, \mathbf{X}, \mathbf{Z}, T, S=0) \cdot f(Y=0, \mathbf{X}, \mathbf{Z}, T, S=1)}{f(Y, \mathbf{X}, \mathbf{Z}, T, S=1) \cdot f(Y=0, \mathbf{X}, \mathbf{Z}, T, S=0)} \\
=& \frac{f(Y, \mathbf{X}, \mathbf{Z}, T, S=0)/f(Y, \mathbf{X}, \mathbf{Z}, T) \cdot f(Y=0, \mathbf{X}, \mathbf{Z}, T, S=1)/f(Y=0, \mathbf{X}, \mathbf{Z}, T)}{f(Y, \mathbf{X}, \mathbf{Z}, T, S=1)/f(Y, \mathbf{X}, \mathbf{Z}, T) \cdot f(Y=0, \mathbf{X}, \mathbf{Z}, T, S=0)/f(Y=0, \mathbf{X}, \mathbf{Z}, T)} \\
=& \frac{f(S=0 \mid \mathbf{X}, \mathbf{Z}, T, Y) \cdot f(S=1 \mid \mathbf{X}, \mathbf{Z}, T, Y=0)}{f(S=0 \mid \mathbf{X}, \mathbf{Z}, T, Y=0) \cdot f(S=1 \mid \mathbf{X}, \mathbf{Z}, T, Y)}
\end{aligned}
$$

Under Assumption 1, because $\mathbf{Z} \perp\!\!\!\perp S \mid \mathbf{X}, T, Y$, the above equation equals $\text{OR}(\mathbf{X}, T, Y)$ in Equation (2). It indicates that the odds ratio function captures the impact of the outcome itself on the sample selection mechanism and is thus a measure of collider bias (Miao & Tchetgen Tchetgen, 2016).

### A.3.2  AN EXPLANATION OF EQUATION (5)

By Equation (2), $\text{OR}(\mathbf{X}, T, Y = 0) = 1$ because

$$\text{OR}(\mathbf{X}, T, Y = 0) = \frac{f(S = 0 \mid \mathbf{X}, T, Y = 0) \cdot f(S = 1 \mid \mathbf{X}, T, Y = 0)}{f(S = 0 \mid \mathbf{X}, T, Y = 0) \cdot f(S = 1 \mid \mathbf{X}, T, Y = 0)}$$
$$= 1.$$

Therefore, by the definition of $\widetilde{\text{OR}}(\mathbf{X}, T, Y)$ that

$$\widetilde{\text{OR}}(\mathbf{X}, T, Y) = \text{OR}(\mathbf{X}, T, Y) / \mathbb{E}[\text{OR}(\mathbf{X}, T, Y) \mid \mathbf{X}, T, S = 1],$$

the right hand side of Equation (5) equals to

$$\frac{\widetilde{\text{OR}}(\mathbf{X}, T, Y)}{\widetilde{\text{OR}}(\mathbf{X}, T, Y = 0)} = \frac{\text{OR}(\mathbf{X}, T, Y) \cdot \mathbb{E}[\text{OR}(\mathbf{X}, T, Y) \mid \mathbf{X}, T, S = 1]}{\text{OR}(\mathbf{X}, T, Y = 0) \cdot \mathbb{E}[\text{OR}(\mathbf{X}, T, Y) \mid \mathbf{X}, T, S = 1]}$$
$$= \frac{\text{OR}(\mathbf{X}, T, Y)}{\text{OR}(\mathbf{X}, T, Y = 0)}$$
$$= \text{OR}(\mathbf{X}, T, Y),$$

which is exactly the left hand side of Equation (5).

### A.3.3  AN EXPLANATION OF $L_{\text{q}}$

As stated in Section 1, ShadowCatcher conducts an additional hypothesis test based on Theorem 2 after the generation phase finishes.

**Theorem 2 (d'Haultfoeuille, 2010).** Suppose the overlap assumption and $\mathbf{Z} \not\perp\!\!\!\perp Y \mid \mathbf{X}, T, S = 1$ hold, then $\mathbf{Z} \perp\!\!\!\perp S \mid \mathbf{X}, T, Y$ can be rejected if and only if there does not exist any function $Q(\cdot)$ that satisfies the following equation and takes value between $(0, 1]$:

$$\mathbb{E}[S/Q(\mathbf{X}, T, Y) - 1 \mid \mathbf{X}, \mathbf{Z}, T] = 0.$$

The tester aims to learn a solution $q$ of $Q(\mathbf{X}, T, Y)$ in Equation (6) that belongs to $(0, 1]$ which turns into an optimization problem by minimizing

$$L_{\text{q}} = \frac{1}{n} \cdot \sum_{i=1}^{n} ||(s_i/q(\mathbf{x}_i, t_i, y_i) - 1) \cdot (\mathbf{x}_i, \mathbf{z}_i, t_i)||_2^2,$$

where $q(\mathbf{x}_i, t_i, y_i)$ is a function from $\mathbb{R}$ to $(0, 1]$ and $|| \cdot ||_2$ denotes the $\ell_2$ norm.

Specifically, if $\mathbb{E}[S/Q(\mathbf{X}, T, Y) - 1 \mid \mathbf{X}, \mathbf{Z}, T] = 0$ (by Theorem 2 in Equation (6)), then

$$\mathbb{E}[\mathbb{E}[S/Q(\mathbf{X}, T, Y) - 1 \mid \mathbf{X}, \mathbf{Z}, T] \cdot (\mathbf{X}, \mathbf{Z}, T)] = 0.$$

The left hand side equals to

$$\mathbb{E}[\mathbb{E}[S/Q(\mathbf{X}, T, Y) - 1 \mid \mathbf{X}, \mathbf{Z}, T] \cdot (\mathbf{X}, \mathbf{Z}, T)] = \mathbb{E}[\mathbb{E}[(S/Q(\mathbf{X}, T, Y) - 1) \cdot (\mathbf{X}, \mathbf{Z}, T) \mid \mathbf{X}, \mathbf{Z}, T]]$$
$$= \mathbb{E}[(S/Q(\mathbf{X}, T, Y) - 1) \cdot (\mathbf{X}, \mathbf{Z}, T)] = 0.$$

Then $L_q$ is just to minimize the square of the $\ell_2$ norm of the last equation.

