# OpenReview forum: "Catch the Shadow: Automatic Shadow Variables Generation for Treatment Effect Estimation under Collider Bias"
_ICLR.cc/2024/Conference — Submitted to ICLR 2024_

### Official Review · Reviewer_Y4SN · 2023-10-17

**Soundness:** 2 fair
**Presentation:** 3 good
**Contribution:** 2 fair
**Rating:** 5
**Confidence:** 3

**Summary:**

This paper proposes a new method to address collider bias in causal effect estimation. The main idea is to use representation learning to generate a shadow variable, in order to alleviate the difficulty in finding a well-defined shadow variable in real-world applcations. The proposed algorithm has three steps which are inspired by exisitng theoretical results on identification using shadow variables, in the first step it generate shadow variable by imposing conditional independence constraints in the learned representation, in the second step it use hypothesis test to ensure the generation is valid, and in the last step it utilize existing CATE estimator for causal effect estimation. Experiments on conducted on IHDP and Twins datasets, and the proposed algorithm is compared with algorithms that are designed to address confounding bias.

After considering the authors' responses and discussions with AC, I have changed my score to 5. My concern is mainly regarding the usage of the Twins dataset, which was used in the original submission, but the authors later decided to remove this dataset from the manuscript.

**Strengths:**

1. Motivation. The motivation of this work is very good. As most work (especially in AI/ML community) on causal effect estimation focus on confounder bias, it is nice to see a work that attempt to address collider bias.
2. Writing. The structure and writing of this work is well-organized and clear.

**Weaknesses:**

1. Insufficient evaluation. The selected baseline methods/datasets are not designed for collider bias evaluation. Please see questions for more details.
2. Limited technical contribution. The proposed method mainly follows the theoretical results outlined by previous works, especially those by Miao and d’Haultfoeuille. Although this is not a critical problem by itself, it is amplified by the fact that the designed algorithm is fragmented into three components. The first and second components (which forms the "Shadow-Catcher"), mandates multiple testing which is subsequently proposed to be addressed by p-value correction. Then the third component comes in and uses existing CATE estimators to obtain the final estimation. There is little to none discussions on the guarantee of the algorithm procedure when putting these three components together.

**Questions:**

1. How is the CATE estimations evaluated on Twins dataset? The Twins dataset is usually used only for evaluating ATE estimations as there is no ground truth counterfactual outcomes.
2. Have the authors compared with more recently proposed CATE/ATE estimators? For example TEDVAE [1], DR-CFR [2]. As the currently compared methods are mainly not designed for collider and are a bit outdated, comparing with more recent methods can further demonstrate if the proposed approach is effective. As we are now at ICLR 2024, it seems insufficient to have most of the baselines proposed in 2016/2017.
3. Also is there any reason for selecting the datasets used in the manuscript? IHDP itself is a semi-synthetic dataset with only one data generation process, which only covers rather limited real-world scenarios. Twins is commonly used for ATE estimation instead of CATE estimation.

[1] Treatment effect estimation with disentangled latent factors. AAAI 2021.

[2] Learning Disentangled Representations for CounterFactual Regression. ICLR 2020.

---

> ### Author Response · Authors · 2023-11-21
> **Responses by Authors (Part 1): Added experiments on the **Jobs** and **ACIC 2016** datasets**
>
> We sincerely appreciate the reviewer’s great efforts and insightful comments to improve our manuscript. In below, we address these concerns point by point and try our best to update the manuscript accordingly.
>
> > **[Q1] How is the CATE estimations evaluated on Twins dataset? The Twins dataset is usually used only for evaluating ATE estimations as there is no ground truth counterfactual outcomes.**
>
> **Response:** The reviewer raises an interesting concern. Following previous studies [1, 2], we use the Twins dataset to evaluate CATE estimation performance. Nonetheless, **we agree with the reviewer that “the Twins dataset is usually used only for evaluating ATE estimations as there is no ground truth counterfactual outcomes”.**
>
> As suggested by the reviewer, **we conducted experiments on two additional datasets with more comprehensive DGPs,** namely **Jobs** and **ACIC 2016**, and **removed CATE evaluation results on Twins dataset** in Table 2. Since the Jobs dataset contains no counterfactual outcomes, we follow [3, 4] to use the policy risk for evaluating the CATE estimation performance, which measures the expected loss if the treatment is taken according to the CATE estimation. The results are shown as below.
>
> | Method | ACIC within-sample | ACIC out-of-sample |Jobs within-sample | Jobs out-of-sample |
> |:---------| :---------: | :---------: | :---------: | :---------: |
> | Heckit | 3.106$\pm$0.444 | 3.340$\pm$0.111 | 0.328$\pm$0.050 | 0.331$\pm$0.052  |
> | DR | 2.346$\pm$0.129 | 2.653$\pm$0.222  | 0.316$\pm$0.007 | 0.317$\pm$0.036 |
> | IPSW    |  4.244$\pm$0.145 | 5.411$\pm$0.073 | 0.284$\pm$0.051  | 0.289$\pm$0.063 |
> | BNN   | 2.173$\pm$0.150 | 2.586$\pm$0.486 | 0.303$\pm$0.025 | 0.304$\pm$0.041|
> | TARNet    | 2.275$\pm$0.756 | 2.805$\pm$0.766 | 0.315$\pm$0.012 | 0.316$\pm$0.050 |
> | CFR   | 2.107$\pm$0.297 | 2.361$\pm$0.587 | 0.313$\pm$0.018 | 0.314$\pm$0.072|
> | CForest     | 4.137$\pm$0.295 | 4.605$\pm$0.137  | 0.326$\pm$0.012 | 0.326$\pm$0.059|
> | DR-CFR     | 2.240$\pm$0.691 | 2.340$\pm$0.663 | 0.322$\pm$0.022 | 0.323$\pm$0.099|
> | TEDVAE     | 3.501$\pm$0.708 | 4.468$\pm$0.813  | 0.296$\pm$0.046 | 0.300$\pm$0.031  |
> | DeR-CFR| 2.214$\pm$0.204| 2.246$\pm$0.598 | 0.309$\pm$0.023 | 0.311$\pm$0.029|
> | DESCN   | 2.185$\pm$0.150 | 2.306$\pm$0.236 | 0.331$\pm$0.010 | 0.331$\pm$0.051 |
> | ES-CFR     | 3.875$\pm$0.224 | 4.494$\pm$0.214   | 0.290$\pm$0.045   | 0.293$\pm$0.046 |
> | Ours (New)    | **1.911$\pm$0.126** | **2.047$\pm$0.351** | **0.279$\pm$0.017** | **0.280$\pm$0.018**|
>
> From the above results, we observe that: (1) The performance of all the estimators is not as good as that on other datasets like the IHDP dataset because the data generation process of the ACIC 2016 datasets is more complex and comprehensive. (2) DR-CFR, DESCN, and ES-CFR show the most competitive performance among all baseline methods. (3) The proposed method stably outperforms all baselines addressing confounding bias or sample selection bias, since the previous methods are limited to the cases in which $X$ and $T$ cause $S$, and cannot handle the cases with addition $Y$ cause $S$. Instead, our ShadowCatcher and ShadowEstimator can address collider bias for accurate CATE estimation. (4) The proposed method on the Jobs dataset shows the lowest policy risk, which demonstrates the effectiveness of our methods in real-world applications.

---

> > ### Author Response · Authors · 2023-11-21
> > **Responses by Authors (Part 2): Added experiments comparing with the most recent CATE estimation methods**
> >
> > > **[Q2] The authors should compare with more recently proposed CATE/ATE estimators.**
> >
> > **Response:** We thank the reviewer for pointing out this issue. Following the reviewer’s suggestion, for both synthetic and real-world experiments, **we added comparisons with more recent CATE estimation baselines**, including: DeR-CFR [1], DR-CFR [5], TEDVAE [6], DESCN [7], and ES-CFR [8]. Among them, **ES-CFR [8] is the state-of-the-art CATE estimation method published in NeurIPS 23,** and the results are as below.
> >
> > | Method | $\beta=1$ selected |  $\beta=1$ unselected |  $\beta=3$ selected | $\beta=3$ unselected | $\beta=5$ selected | $\beta=5$ unselected |
> > |:---------| :---------: | :---------: | :---------: | :---------: | :---------: | :---------: |
> > DeR-CFR | 0.291$\pm$0.010 | 0.309$\pm$0.014 | 0.323$\pm$0.015 | 0.348$\pm$0.017 | 0.358$\pm$0.011 | 0.439$\pm$0.013|
> > DR-CFR  | 0.284$\pm$0.038 | 0.307$\pm$0.040 | 0.340$\pm$0.055 | 0.355$\pm$0.064 | 0.366$\pm$0.051 | 0.435$\pm$0.060|
> > TEDVAE  | 0.281$\pm$0.056 | 0.419$\pm$0.070 | 0.378$\pm$0.063 | 0.420$\pm$0.059 | 0.394$\pm$0.054 | 0.431$\pm$0.067|
> > DESCN | 0.295$\pm$0.002 | 0.312$\pm$0.002 | 0.326$\pm$0.003 | 0.357$\pm$0.004 | 0.365$\pm$0.003 | 0.449$\pm$0.011|
> > ES-CFR  | 0.289$\pm$0.003 | 0.305$\pm$0.004 | 0.331$\pm$0.003 | 0.359$\pm$0.003 | 0.369$\pm$0.003 | 0.448$\pm$0.005|
> > Ours (New)    | **0.227$\pm$0.001** | **0.229$\pm$0.001** | **0.249$\pm$0.013** | **0.255$\pm$0.021** | **0.299$\pm$0.008** | **0.300$\pm$0.008** |
> > |||||||
> >
> > | Method | IHDP within-sample |  IHDP out-of-sample |  ACIC within-sample | ACIC out-of-sample | Jobs within-sample | Jobs out-of-sample |
> > |:---------| :---------: | :---------: | :---------: | :---------: | :---------: | :---------: |
> > |DR-CFR |  1.299$\pm$0.087 | 1.399$\pm$0.171 | 2.240$\pm$0.691 | 2.340$\pm$0.663 | 0.322$\pm$0.022 | 0.323$\pm$0.099|
> > |TEDVAE     | 4.246$\pm$0.394 | 4.347$\pm$0.563 | 3.501$\pm$0.708 | 4.468$\pm$0.813  | 0.296$\pm$0.046 | 0.300$\pm$0.031  |
> > |DeR-CFR | 1.446$\pm$0.345 | 1.571$\pm$0.371 | 2.214$\pm$0.204 | 2.246$\pm$0.598  | 0.309$\pm$0.023 | 0.311$\pm$0.029|
> > |DESCN   | 1.193$\pm$0.057 | 1.665$\pm$0.246 | 2.185$\pm$0.150 | 2.306$\pm$0.236 | 0.331$\pm$0.010 | 0.331$\pm$0.051 |
> > |ES-CFR     | 1.499$\pm$0.096 | 1.436$\pm$0.095 | 3.875$\pm$0.224 | 4.494$\pm$0.214   | 0.290$\pm$0.045   | 0.293$\pm$0.046 |
> > |Ours (New)    | **0.703$\pm$0.106** | **0.723$\pm$0.102** | **1.911$\pm$0.126** | **2.047$\pm$0.351** | **0.279$\pm$0.017** | **0.280$\pm$0.018** |
> > ||||||||
> >
> > The results show that our method outperforms all these more recent CATE estimation baselines, demonstrating the effectiveness of our method.
> >
> > > **[Q3] Is there any reason for selecting the datasets used in the manuscript? IHDP itself is a semi-synthetic dataset with only one data generation process, which only covers rather limited real-world scenarios. Twins is commonly used for ATE estimation instead of CATE estimation.**
> >
> > **Response:** Thank you for the comment.
> >
> > -	Since IHDP is a semi-synthetic dataset with only one data generation process, **we added experiments on **ACIC 2016** dataset with more comprehensive DGPs.**
> > -	Since Twins is commonly used for ATE estimation instead of CATE estimation, **we added experiments on a new real-world dataset,** namely the **Jobs** dataset, and **removed CATE evaluation results on Twins dataset** in Table 2.
> >
> > Please kindly refer to our response to [Q1] and Table 2 in the revised manuscript.
> >
> > > **[W1] The selected baseline methods/datasets are not designed for collider bias evaluation.**
> >
> > **Response:** We thank the reviewer for the useful comments.
> >
> > -	For **selected datasets,** to the best of our knowledge, there are no public datasets with CATE ground truths designed for collider bias, thus we choose the widely-used datasets for evaluating the CATE estimation and manually introduce the varying levels of collider bias.
> > -	For **selected baseline methods,** to the best of our knowledge, we are the first approach to adopt representation learning for addressing the collider bias in CATE estimation (please kindly correct us if we were wrong). For the previous statistical methods tackling the collider bias, they all need prior knowledge of instrumental variables or shadow variables, resulting in different data forms.
> >
> > Therefore, we compare our method with the baseline methods addressing confounding bias or sample selection bias in which $X$ and $T$ cause $S$, but not $Y$ cause $S$.

---

> > > ### Author Response · Authors · 2023-11-21
> > > **Responses by Authors (Part 3): More discussions on the guarantee of the algorithm procedure**
> > >
> > > > **[W2] There is little to none discussions on the guarantee of the algorithm procedure when putting these three components together.**
> > >
> > > **Response:** We thank the reviewer for pointing out this issue. We also notice that Reviewer 5YRg considered the proposed methods a bit hacky (a combination of multiple steps), but all the steps are well-motivated. To make the entire framework of ShadowCatcher and ShadowEstimator more connected and clearer, **we briefly summarize the overall process** of using ShadowCatcher and ShadowEstimator:
> > >
> > > -	ShadowCatcher first takes the fully observed $X$ and $T$, and the observed $Y$ of $S=1$ units as inputs to generate shadow-variable representations $Z$, and then tests whether the generated $Z$ satisfies Assumption 1.
> > > -	If the generated $Z$ does not pass the hypothesis test, ShadowCatcher should re-generate new shadow-variable representations until the generated $Z$ finally passes the test.
> > > -	Finally, ShadowEstimator uses the generated $Z$ to estimate treatment effects with observational samples.
> > >
> > > For a clearer presentation, **we add an overall flowchart as shown in Figure 4 on page 14.**
> > >
> > > Moreover, **we have the following discussions on the guarantee of the algorithm procedure when putting these three components together,** including: (1) the generation phase of ShadowCatcher, (2) the hypothesis test phase of ShadowCatcher, and (3) ShadowEstimator.
> > >
> > > - Putting together (1) the generation phase of ShadowCatcher and (2) the hypothesis test phase of ShadowCatcher is reasonable: as we wrote in Section 3.3, the hypothesis test is theoretically guaranteed (by Theorem 2) to be able to reject $Z$ that does not satisfy the shadow-variable assumption. Therefore, similar to the standard statistical process for finding the shadow variables with the validity hypothesis test, putting together (1) and (2) has sound theoretical guarantees, and one of our main contributions is the automatically generated $Z$.
> > >
> > > - Putting together (1, 2) and (3) ShadowEstimator is also reasonable: as we wrote in Section 3.2, the valid $Z$ makes $f(Y | X,T,Z,S=0)$ identifiable, which is theoretically guaranteed (by Theorem 1). Since the generated $Z$ by ShadowCatcher passes the hypothesis test, as well as the ShadowEstimator strictly follows the theoretical results of using $Z$ to identify $f(Y | X,T,Z,S=0)$, the whole procedure has theoretical guarantees following the well-established shadow variable theory.
> > >
> > > ***
> > >
> > > **We hope the above discussion will fully address your concerns about our work, and we would really appreciate it if you could be generous in raising your score.** We look forward to your insightful and constructive responses to further help us improve the quality of our work. Thank you!
> > >
> > > ***
> > >
> > > > **References**
> > >
> > > [1] Wu, A., et al. Learning Decomposed Representations for Treatment Effect Estimation. TKDE 2022.
> > >
> > > [2] Jinsung, Y., et al. GANITE: Estimation of Individualized Treatment Effects using Generative Adversarial Nets. ICLR 2018.
> > >
> > > [3] Shalit, U., et al. Estimating individual treatment effect: generalization bounds and algorithms. ICML 2017.
> > >
> > > [4] Louizos, C., et al. Causal Effect Inference with Deep Latent-Variable Models. NIPS 2017.
> > >
> > > [5] Greiner, N. H. R. Learning Disentangled Representations for CounterFactual Regression. ICLR 2020.
> > >
> > > [6] Zhang, W., et al. Treatment Effect Estimation with Disentangled Latent Factors. AAAI 2021.
> > >
> > > [7] Zhong, K., et al.  DESCN: Deep Entire Space Cross Networks for Individual Treatment Effect Estimation. KDD 2022.
> > >
> > > [8] Hao, W., et al. Optimal Transport for Treatment Effect Estimation. NeurIPS 2023.

---

> > ### Comment · Reviewer_Y4SN · 2023-11-21
> >
> > I thank the authors for their effort in the rebuttal. These addressed most of my concerns. As there is not much time left in the author-reviewer discussion period, I will only ask two follow-up questions.
> >
> > 1. When using the IHDP, ACIC semi-synthetic datasets, have the authors examined the different settings' data-generating process (DGP)? Do these DGPs actually induce collider bias? If so, roughly what proportion of these DGPs induce collider bias? If not, what is the reason behind the performance advantages gained by the proposed? These questions would shed more light on the performance gain. The DGPs of these datasets are readily available on Github.
> >
> > 2. Regarding removing the Twins dataset results, can you give more details on how the evaluation was done in the original submission?

---

> > > ### Author Response · Authors · 2023-11-22
> > > **Responses by Authors (Part 4): Responses to the two follow-up questions**
> > >
> > > We sincerely appreciate the reviewer’s timely response and further discussions. In below, we try our best to address your new concerns point by point.
> > >
> > >
> > > > **[Q1] When using the IHDP, ACIC semi-synthetic datasets, have the authors examined the different settings' data-generating process (DGP)? Do these DGPs actually induce collider bias? If so, roughly what proportion of these DGPs induce collider bias? If not, what is the reason behind the performance advantages gained by the proposed?**
> > >
> > > **Response:** We thank the reviewer for pointing out this issue. **Yes, we have examined the different settings' DGPs in the semi-synthetic datasets.** Please kindly refer to the experimental results in our previous responses.
> > >
> > > **However, it is not the DAG itself that causes collider bias, but rather we manually introduced collider bias for different settings' DGPs.** Notably, **the different settings' DGP in the IHDP and ACIC datasets are limited to between $X$, $T$, and $Y$, but not $S$.** Collider bias is essentially a sample selection mechanism, where all $X$, $T$, and $Y$ affect the sample selection indicator $S$. For samples with $S=1$, all $X$, $T$, and $Y$ are observed, whereas for samples with $S=0$, only $X$ and $T$ are observed. To the best of our knowledge, there are no public available dataset with such a data format corresponding to the above collider bias, thus **the original DGPs of the datasets do not induce any collider bias** and we manually introduce the varying levels of collider bias to different settings' DGPs.
> > >
> > > - To introduce collider bias into the ACIC 2016 dataset, we simulate $S$ detailed in Appendix A.2.3: $S\sim \operatorname{Bernoulli}(1/(1+e^{-s(\mathbf{X}, T)}))$, where $s(\mathbf{X}, T) = T - Y  + \sum_{i=1}^{d_{\mathrm{s}}}(\mathbf{1}(\operatorname{mod}(i,2)\equiv 1) - \mathbf{1}(\operatorname{mod}(i,2)\neq 1)) \cdot X_i / d) + \epsilon_{\mathrm{s}}$ with $\epsilon_{\mathrm{s}}\sim \mathcal{N}(0,1)$ and $d=58$.
> > >
> > > - For the IHDP dataset, which is from a study evaluating the effect of specialist home visits ($T$) on the future cognitive test scores of premature infants ($Y$), we introduce collider bias in the following way that is more applicable to the real-world scenarios. Specifically, we set $S=0$ for units in the control group ($T=0$) whose mother has boozing problem (one binary covariate is 1) and the infant's cognitive test score ($Y$) is lower than the mean value. Intuitively, unlike the treated group, which can carefully design and regularly follow up to ensure the collection of effective test results, the control group is more likely to have sample selection bias. For those mothers with boozing problems and mothers whose children have weaker cognitive abilities, it is more likely that they will not take their children to participate in the cognitive test, resulting in collider bias in real-world scenarios.
> > >
> > > Therefore, the datasets we use are collider-biased, from which our method demonstrates significantly superior performance in both datasets. Please kindly refer to Appendix A.2.3 for more collider bias details.
> > >
> > >
> > > > **[Q2] Regarding removing the Twins dataset results, can you give more details on how the evaluation was done in the original submission?**
> > >
> > > **Response:** Thank you for the comment. Following previous works [1, 2, 3, 4] using the Twins dataset for evaluating CATE estimation, we define the treatment $t=1$ as being the heavier twin (and $t=0$ as being the lighter twin). The outcome is defined as the 1-year mortality. **In this setting, for each twin pair we observed both the case $t=0$ (lighter twin) and $t=1$ (heavier twin). Thus, the ground truth of individualized treatment effect is known in this dataset.** In order to simulate an observational study for CATE estimation, we selectively observe one of the two twins using the feature information (creating selection bias) as follows: $t \mid \mathbf{x} \sim \operatorname{Bern}\left(\operatorname{Sigmoid}\left(\mathbf{w}^T \mathbf{x}+n\right)\right)$ where $\mathbf{w}^T \sim \mathcal{U}\left((-0.1,0.1)^{30 \times 1}\right)$ and $n \sim \mathcal{N}(0,0.1)$.
> > >
> > > ***
> > >
> > > **We sincerely hope that our further response above can address your remaining concerns well, and we are more than eager to have further follow-up discussions.**
> > >
> > > ***
> > >
> > > > **References**
> > >
> > > [1] Louizos, C., et al. Causal Effect Inference with Deep Latent-Variable Models. NeurIPS 2017.
> > >
> > > [2] Jinsung, Y., et al. GANITE: Estimation of Individualized Treatment Effects using Generative Adversarial Nets. ICLR 2018.
> > >
> > > [3] Liuyi Y, et al. Representation Learning for Treatment Effect Estimation from Observational Data. NeurIPS 2018
> > >
> > > [4] Alicia C, et al. On Inductive Biases for Heterogeneous Treatment Effect Estimation. NeurIPS 2021.

---

> > > > ### Author Response · Authors · 2023-11-23
> > > > **We are still expecting to have your response in the final author-reviewer discussion phase!**
> > > >
> > > > Dear Reviewer Y4SN,
> > > >
> > > > We sincerely appreciate your time and efforts in the review and discussion phases. We have tried our best to respond to all your questions and concerns in detail. **As the author-reviewer discussion phase is only one hour left, if our responses have addressed your remaining concerns, we would really appreciate it if you could be generous in raising your score**; if not, we look forward to your further insightful and constructive responses to further help us improve the quality of our work. Thank you!
> > > >
> > > > Best regards,
> > > >
> > > > Submission5512 Authors.

---

### Official Review · Reviewer_5YRg · 2023-10-26

**Soundness:** 4 excellent
**Presentation:** 3 good
**Contribution:** 4 excellent
**Rating:** 8
**Confidence:** 4

**Summary:**

POST-REBUTTAL UPDATE

Thanks for the clarifications, which addressed my concerns.

POST-REBUTTAL UPDATE END

The paper considers the problem of causal estimation with collider bias, that is, in situations where whether the individual is sampled or not (S=1/0) depends on the treatment T and the outcome Y. Previously, it has been shown that causal estimation can be done when there is collider bias if there exists a “shadow variable” (Z), which depends on the outcome variable (Y) among the sampled individuals (S=1), conditionally on the treatment (T) and covariates (X). Notably, Y can be missing for individuals that do not belong in the sample (S=0). Additionally, the shadow variable Z should be independent of the selection indicator S conditionally on all other variables. The main problem in using shadow variables is that they don’t always exist. The innovation of the paper is to learn suitable shadow variables from the observed covariates (X).

**Strengths:**

1) Most of the prior works addressing bias in causal estimation have focused on resolving the problem of hidden confounders and little work exists on collider bias. Therefore, the topic is fresh and important, and I found the approach based on shadow variables interesting.
2) The method seems technically solid and clearly presented. Rationale of the different steps of the method are justified properly.
3) The empirical validation seems appropriate, containing multiple reasonable baselines, two real-world datasets, and simulations.

**Weaknesses:**

I did not identify any major weaknesses. Some smaller ones:
1) The assumption that X and T are observed when the individual is not sampled seems quite strong and probably not true in many real-world use cases (see Question 1 below to address this).
2) Overall, the the method appears a bit hacky (a combination of multiple steps), but all the steps are well-motivated.
3) When printed out, fonts in Figures 2 and 3 are barely readable.

**Questions:**

1) I guess quite often if the individual is not sampled, it’s not only Y that is not observed but also X (and T), in which case the shadow variable approach would not be applicable. Could the authors discuss how this affects the usefulness in practice, and give representative real-world examples about situations in which they expect the method to be useful and where not?

2) I don’t understand the equation for the loss function in the “Hypothesis Test Phase” paragraph. Could there be a typo?

---

> ### Author Response · Authors · 2023-11-21
> **Responses by Authors: More discussions on real-world use cases and formula correction**
>
> We sincerely appreciate the reviewer’s great efforts and insightful comments to improve our manuscript. In below, we address these concerns point by point and try our best to update the manuscript accordingly.
>
> > **[W1 & Q1] If the individual is not sampled, it’s not only Y that is not observed but also X (and T), in which case the shadow variable approach would not be applicable. How does this affect the usefulness in practice. Are there any representative real-world examples about situations in which they expect the method to be useful and where not?**
>
> **Response:** Thank you for the comment. We agree with the reviewer that there are also cases "if the individual is not sampled, it's not only $Y$ that is not observed but also $X$ (and $T$), in which case the shadow variable approach would not be applicable". **Instead, the proposed method (and most methods on sample selection bias) is applicable when the target population is determined, which also covers a wide range of real-world scenarios.**
>
> For example, when studying the effect of vaccines ($T$) on disease prevention ($Y$), the target population is determined as, for example, the population of a country or region. Given the target population, governments or institutions would collect all information in terms of their covariates ($X$), and vaccine status ($T$). However, only a portion of the population will remain in follow-up ($S=1$), making the outcome ($Y$) able to be collected, while others will be lost ($S=0$), due to emigration, death, etc.
>
> Another example is semi-supervised learning, in which one of the covariates is regarded as the treatment ($T$), and the rest of the covariates are denoted as $X$. Then the labeled samples are marked as $S=1$ with fully observed $X$, $T$, and $Y$, whereas the unlabeled samples are marked as $S=0$ with only $X$ and $T$ observed.
>
> > **[W2] Overall, the method appears a bit hacky (a combination of multiple steps), but all the steps are well-motivated.**
>
> **Response:** Thank you for the comment. We agree that the proposed method is a bit hacky but well-motivated. To make the entire framework of ShadowCatcher and ShadowEstimator more connected and clearer, **we briefly summarize the overall process** of using ShadowCatcher and ShadowEstimator:
>
> -	ShadowCatcher first takes the fully observed $X$ and $T$, and the observed $Y$ of $S=1$ units as inputs to generate shadow-variable representations $Z$, and then tests whether the generated $Z$ satisfies Assumption 1.
> -	If the generated $Z$ does not pass the hypothesis test, ShadowCatcher should re-generate new shadow-variable representations until the generated $Z$ finally passes the test.
> -	Finally, ShadowEstimator uses the generated $Z$ to estimate treatment effects with observational samples.
>
> **We also add an overall flowchart as shown in Figure 4 on page 14.**
>
> > **[W3] When printed out, fonts in Figures 2 and 3 are barely readable.**
>
> **Response:** We thank the reviewer for pointing out this issue. **We have adjusted them in our revised version** on page 5 and Figure 3 on page 7 to make them clearer.
>
> > **[Q2] There is a typo in the equation for the loss function in the “Hypothesis Test Phase” paragraph.**
>
> **Response:** We thank the reviewer for pointing out typos here. **The correct equation should be $L_q = 1/n * \sum_{i=1}^n||(s_i / q(x_i, t_i, y_i) - 1) \cdot (x_i, z_i, t_i)||_2^2$, where $(x_i, z_i, t_i)$ is a vector and $||.||_2$ is the L2-norm.** $L_q$ aims to learn a solution q of Q by Theorem 2 in Equation (6). Specifically, if $E[S / Q(X, T, Y) - 1 | X, Z, T] = 0$ (by Theorem 2 in Equation (6)), then $E[E[S / Q(X, T, Y) - 1 | X, Z, T] * (X, Z, T) ]=0$. The left hand side equals to $E[(S / Q(X, T, Y) - 1) * (X, Z, T)] = 0$. Then $L_q$ is just to minimize the square of the L2-norm of that Equation. We have corrected that on page 6, and added a more detailed explanation of $L_q$ in Appendix A.3.3 on page 18.
>
> ***
>
> **We hope the above discussion will fully address your concerns about our work.** We look forward to your insightful and constructive responses to further help us improve the quality of our work. Thank you!

---

### Official Review · Reviewer_G11c · 2023-10-30

**Soundness:** 3 good
**Presentation:** 3 good
**Contribution:** 4 excellent
**Rating:** 8
**Confidence:** 4

**Summary:**

In this paper, the authors focus on the collider bias problem, which is one of the important challenges of causal inference. They propose a novel method that can automatically generate shadow variable representations from observed covariates and propose an estimator to estimate CATE with the help of the generated representations. They conduct extensive experiments, including comparing different choices of the hyper-parameter $alpha$ and ablation studies. The main contribution is that they relax the strong assumptions of previous works on collider bias and make CATE estimation under collider bias feasible in most real-world observational studies.

**Strengths:**

1. Collider bias is an important and easily ignored problem of causal inference in observational studies. The main difficulty for previous works to be applied in real-world scenarios is the strong assumptions they made. Therefore, if the assumptions are relaxed, the proposed method will make significant contributions to the causality community and will have high application value in real-world scenarios.
2. This paper is mainly based on the shadow variables identification framework of collider bias, which strongly assumes that valid shadow variables are well-defined. Interestingly, the authors propose a novel idea that the shadow-variable representations can be learned from the observed covariates and propose a novel ShadowCatcher to "catch the shadow" from the covariates. The success of ShadowCatcher significantly relaxes the strong assumptions of previous works and makes CATE estimation possible under collider bias with the help of the proposed ShadowEstimator. This contributes a lot to causal inference research because, finally, collider bias can be addressed without any strong and even untestable assumptions as confounding bias in observational studies.
3. The proposed method is clearly stated and reasonable to me. For the testable conditional dependence assumption of shadow variables, they directly constrain it in the representation learning phase, and for the untestable conditional independence assumption, they constrain it and do additional hypothesis tests to guarantee it. The entire learning process of ShadowCatcher is theoretically feasible. ShadowEstimator is based on the shadow variables estimation framework, whose correctness is also theoretically guaranteed. There is still one concern about ShadowCatcher, as I will state in the weaknesses part.
4. The experiments are detailed and persuasive. The authors conduct experiments under different strengths of collider bias, proving the ability to reduce collider bias of their methods. They also conduct ablations to prove the effectiveness of the conditional dependence constraint in ShadowCatcher and compare the performance and efficiency under different choices of the hyper-parameter $alpha$. The results and analysis seem reasonable to me.

**Weaknesses:**

1. Since the conditional independence assumption is not strictly constrained due to missing data, the authors use a hypothesis test phase to test whether this assumption is satisfied and only the generated representations pass the test can ShadowCatcher finish learning. As the authors also state in the paper, the choice of $alpha$ will significantly affect the efficiency of ShadowCatcher. As the results in Table 5 and Figure 4 show, the smaller $alpha$ is (which means the test is more strict), the better the performance is. But the number of iterations also gets bigger when the test is too strict. Therefore, the tradeoff between the efficiency and performance of the proposed method should be considered carefully in real-world applications.

2. A minor concern is that the figure size is somewhat small.

**Questions:**

See weakness

---

> ### Author Response · Authors · 2023-11-21
> **Responses by Authors: Experiments and discussions in terms of trade-off between the efficiency and performance**
>
> We sincerely appreciate the reviewer’s great efforts and insightful comments to improve our manuscript. In below, we address these concerns point by point and try our best to update the manuscript accordingly.
>
> > **[W1] The tradeoff between the efficiency and performance of the proposed method should be considered carefully in real-world applications.**
>
> **Response:** Thank you for your constructive suggestion. **We have conducted additional experiments to explore the impact of different choices of $\alpha$ on the efficiency and performance**. The results are shown as below.
>
> | Reject threshold | $\sqrt{\mathrm{PEHE}}$ selected data |  $\sqrt{\mathrm{PEHE}}$ unselected data | mean number of iterations |
> |:---------| :---------: | :---------: | :---------: |
> |$1e-4$| 0.235$\pm$0.003 | 0.235$\pm$0.008 | **1.2** |
> | $5e-5$| 0.228$\pm$0.003 | 0.231$\pm$0.004 | 2 |
> | $1e-5$| 0.228$\pm$0.002 | 0.231$\pm$0.001 | 4.6 |
> | $5e-6$| 0.227$\pm$0.002 | 0.230$\pm$0.002 | 7.9 |
> | $1e-6$| **0.227$\pm$0.001** | **0.229$\pm$0.001** | 10 |
> | | | | |
>
> The above results show that the performance of ShadowCatcher improves as the reject threshold decreases, because the hypothesis test gets more strict, and the constraint gets more reliable. However, the number of iterations required for ShadowCatcher to pass the hypothesis test also increases rapidly, reducing the efficiency of ShadowCatcher. Therefore, choosing an appropriate $\alpha$ is a trade-off between efficiency and performance, which depends on the real-world application scenarios.
>
> > **[W2] The figure size is somewhat small.**
>
> **Response:** We thank the reviewer for pointing out this issue. **We have adjusted them in our revised version on page 5 and Figure 3 on page 7 to make them clearer.**
>
> ***
>
> **We hope the above discussion will fully address your concerns about our work.** We look forward to your insightful and constructive responses to further help us improve the quality of our work. Thank you!

---

### Official Review · Reviewer_JJeK · 2023-10-30

**Soundness:** 4 excellent
**Presentation:** 4 excellent
**Contribution:** 4 excellent
**Rating:** 6
**Confidence:** 3

**Summary:**

This paper presents an approach for learning representations that can take the role of "shadow variables", which allow for CATE identification in the presence of selection bias. They describe the process of learning these variables and empirically show with synthetic and semi-synthetic data that they produce more accurate CATE estimates.

**Strengths:**

- clean and clear contribution, seems like an inventive and well motivated approach for applying ML tools to improve causal inference with good grounding in causal literature
- paper is mostly well presented
- experiments seem to support the main idea of the paper and contrast with other causal methods for slightly different tasks, showing improvement

**Weaknesses:**

- I found myself getting just a little lost in the preliminaries of the shadow variable, particularly at the bottom of page 4. I found it confusing to say that f(Z | X, S) was identifiable from the observed data, and then to still say that we needed to find \tilde(OR) - I thought the f(Z|...) functions eliminated the need to calculate \tilde(OR) according to equation 4. I also don't quite see why Eq 5 is true and think this could use more explanation.
- I found some of the loss functions through 3.3 and 3.4 a little unintuitive. 1) I found it a little odd to try to look at -Z to maximize MSE - this means that the function will behave differently for Z close to 0 since Z and -Z are near each other. Would it be reasonable or more sensible to take random Z instead? or why is -Z the best idea? 2) I'm not sure why h_r, a function that aims to predict Z well, will also help to move the Z0 and Z1 distributions towards each other. 3) what is the notation \dot (x_i, z_i, t_i) in the loss function for Q? it's not clear if there's a typo here, since this just looks like a tuple. 4) I'm not sure why distilling h_z0 / h_z1 into a \tilde(or) function is necessary
- I'm confused by the comment right at the end of 3 around deconfounding methods: I thought there was an assumption around unconfoundedness. Is this still a fair comparison to other methods if deconfounding methods are used?
- In the synthetic data in Table 1, I was surprised that some of the wins were not that big. Given that this is fully synthetic data presented for this method, I'd expect the results to be outside the confidence bands, but in a few spots they're highly overlapping - it makes me wonder if some of the practical choices aren't as effective
- there's a lot of experimental info missing from section 4 around what learning algorithm and models are used


Smaller comments:
- end of Sec 3.3: it says "the final generated Z that passes the test" - should this be the first generated Z which passes the test? it seems like that test is a "stopping criteria" to me

**Questions:**

- What are the failure modes of this method - under what circumstances will learning a shadow variable be harder than others?

---

> ### Author Response · Authors · 2023-11-21
> **Responses by Authors (Part 1): Thanks for your insightful suggestion to take random $Z$ instead, which results in significant performance improvements**
>
> We sincerely appreciate the reviewer’s great efforts and insightful comments to improve our manuscript. In below, we address these concerns point by point and try our best to update the manuscript accordingly.
>
> > **[W1.1] Why do the $f(Z|...)$ functions not eliminate the need to calculate $\widetilde{OR}$ according to Equation (4)?**
>
> **Response:** Thank you for the comment. The necessity for further calculating $\widetilde{OR}(X, T, Y)$ given $f(Z | X, T, S)$ comes from that: for identification of $f(Y | X, T, S=0)$, we need to obtain **both $OR(X, T, Y)$ and $E[\widetilde{OR}(X, T, Y) | X, Z, T, S=1]$** by Equation (2). The latter can be easily calculated by Equation (4) using $f(Z | X, T, S)$. However, **the former still needs $\widetilde{OR}(X, T, Y)$ and $\widetilde{OR}(X, T, Y=0)$** by Equation (5). Thus, it is still necessary to know what $\widetilde{OR}(X, T, Y)$ as a function of $X, T, Y$ is to calculate $\widetilde{OR}(X, T, Y=0)$, and we need to **further solve for $\widetilde{OR}(X, T, Y)$ as a function of $X, T, Y$ from the calculated $E[\widetilde{OR}(X, T, Y) | X, Z, T, S=1]$** by Equation (4).
>
> > **[W1.2] Why is Equation (5) true?**
>
> **Response:** By Equation (2), $OR(X, T, Y=0) = 1$ (just substitute $Y=0$ into Equation (2)). Therefore, by the definition of $\widetilde{OR}(X, T, Y)$ that $\widetilde {OR}(X, T, Y) = OR(X, T, Y) / E[OR(X, T, Y) | X, T, S=1]$, **the right-hand side of Equation (5) equals to $OR(X, T, Y) / OR(X, T, Y=0)$**. Since $OR(X, T, Y=0) = 1$, we have **$OR(X, T, Y) / OR(X, T, Y=0)$ equals the left-hand side of Equation (5) (i.e., $OR(X, T, Y)$)**. To make this clearer, **we make a detailed explanation of Equation (5) in Appendix A.3.2 on page 18**.
>
> > **[W2.1] Would it be reasonable or more sensible to take random $Z$ as the implementation of $Z^-$ instead?**
>
> **Response:** Thank you for such insightful advice. **It is absolutely reasonable and we have incorporated this suggestion throughout our paper.** In fact, the original description of $Z^-$ in our article that "the opposite value of $Z$" is not accurate since we misuse the word "opposite." **We have corrected it in our revised version (on Page 6) that $Z^-$ indeed denotes a value that differs from $Z$ to constrain the conditional dependence assumption**, which any strategy can implement. We choose $-Z$ as $Z^-$ for continuous $Z$s just because this is easy to implement. We do agree that **taking a random Z instead is a better idea** since when $Z$ is close to 0, $-Z$ does not satisfy the requirement of $Z^-$. **We use this new implementation to re-run all experiments and achieve obvious performance improvements**. The results are shown as below.
>
> | Method | $\beta=1$ selected |  $\beta=1$ unselected |  $\beta=3$ selected | $\beta=3$ unselected | $\beta=5$ selected | $\beta=5$ unselected |
> |:---------| :---------: | :---------: | :---------: | :---------: | :---------: | :---------: |
> | Ours (Old) | 0.241$\pm$0.014 | 0.248$\pm$0.009 | 0.305$\pm$0.013 | 0.326$\pm$0.015 | 0.333$\pm$0.040 | 0.404$\pm$0.053 |
> | Ours (New) | **0.227$\pm$0.001** | **0.229$\pm$0.001** | **0.249$\pm$0.013** | **0.255$\pm$0.021** | **0.299$\pm$0.008** | **0.300$\pm$0.008** |
> ||||||||
>
> | Method | IHDP within-sample |  IHDP out-of-sample |  ACIC within-sample | ACIC out-of-sample | Jobs within-sample | Jobs out-of-sample |
> |:---------| :---------: | :---------: | :---------: | :---------: | :---------: | :---------: |
> | Ours (Old) | 1.039$\pm$0.069 | 1.065$\pm$0.099 | 2.078$\pm$0.333 | 2.142$\pm$0.390 | 0.283$\pm$0.018 | 0.284$\pm$0.080 |
> | Ours (New) | **0.703$\pm$0.106** | **0.723$\pm$0.102** | **1.911$\pm$0.126** | **2.047$\pm$0.351** | **0.279$\pm$0.017** | **0.280$\pm$0.018** |
> ||||||||
>
> The original implementation of our method is denoted as Ours (Old), whereas the new implementation of our method is denoted as Ours (New). Note that we add an additional real-world dataset - Jobs to better evaluate the performance, where the evaluation metric is the policy risk $\hat{R}_{\mathrm{Pol}}=1-(\mathbb{E}[Y(1)\mid \tau(\mathbf{x}) > 0,T=1] \cdot \mathbb{P}(\tau(\mathbf{x}) > 0) + \mathbb{E}[Y(0)\mid \tau(\mathbf{x}) \leq 0,T=0]\cdot \mathbb{P}(\tau(\mathbf{x}) \leq 0)$ instead of $\sqrt{\mathrm{PEHE}}$ for other datasets. **The new implementation (taking a random $Z$ instead of $-Z$ as $Z^{-}$) shows obvious performance improvements on all datasets.** We really appreciate your great advice!
>
> > **[W2.2] Why will $h_r$, a function that aims to predict Z well, also help to move the $Z_0$ and $Z_1$ distributions towards each other?**
>
> **Response:** The reviewer raises an interesting concern. Please kindly note that $h_r$ itself is not the key to moving the $Z_0$ and $Z_1$ distributions towards each other. Instead, **as $h_r$ is the estimate of $f(Z | X, T, Y, S=1)$, if $h_r$ can also predict $Z$ well for $S=0$ data, then $f(Z | X, T, Y, S=1) = f(Z | X, T, Y, S=0)$** and the assumption that $Z$ is independent of $S$ given $X, T, Y$ is satisfied.

---

> ### Author Response · Authors · 2023-11-21
> **Responses by Authors (Part 2): Notation correction and missing experimental details**
>
> > **[W2.3] What is the notation $\cdot (x_i, z_i, t_i)$ in the loss function for $Q$? it's not clear if there's a typo here.**
>
> **Response:** We thank the reviewer for pointing out typos here. **The correct equation should be $L_q = 1/n * \sum_{i=1}^n||(s_i / q(x_i, t_i, y_i) - 1) \cdot (x_i, z_i, t_i)||_2^2$, where $(x_i, z_i, t_i)$ is a vector and $||.||_2$ is the L2-norm.** $L_q$ aims to learn a solution q of Q by Theorem 2 in Equation (6). Specifically, if $E[S / Q(X, T, Y) - 1 | X, Z, T] = 0$ (by Theorem 2 in Equation (6)), then $E[E[S / Q(X, T, Y) - 1 | X, Z, T] * (X, Z, T) ]=0$. The left hand side equals to $E[(S / Q(X, T, Y) - 1) * (X, Z, T)] = 0$. Then $L_q$ is just to minimize the square of the L2-norm of that Equation. We have corrected that on page 6, and added a more detailed explanation of $L_q$ in Appendix A.3.3 on page 18.
>
> > **[W2.4] why is distilling $h_{z_0} / h_{z_1}$ into a $\widetilde{or}$ function is necessary?**
>
> **Response:** Thank you for the comment, but we believe this question comes from [W1.1] in which Proposition 1 and Equation (5) confuse you. Please kindly refer to our **response to [W1.1]**, it is necessary because we need to **further solve for $\widetilde{OR}(X, T, Y)$ as a function of $X, T, Y$ from the calculated $E[\widetilde{OR}(X, T, Y) | X, Z, T, S=1]$ by Equation (4)**, so that we can obtain $OR(X, T, Y)$ from $\widetilde{OR}(X, T, Y)$ by Equation (5) and further identify $f(Y | X, Z, T, S=0)$ by Equation (2).
>
> > **[W3] I thought there was an assumption around unconfoundedness. Is this still a fair comparison to other methods if de-confounding methods are used?**
>
> **Response:** The reviewer raises an interesting concern. However, the de-confounding way is just an implementation of the outcome estimators in ShadowCatcher and ShadowEstimator, which **will not affect (either positive or negative) the ability to address collider bias**. As you may noticed, we do need the unconfoundedness assumption on the target population (but not on only $S=1$ data because this assumption can be violated due to collider bias). This means we assume that there are no unobserved confounders, but there can possibly be confounding bias caused by fully observed covariates. The de-confounding method (an IPM to balance treated and controlled representations following [1]) we use is just to **avoid the impact of such confounding bias on CATE estimation, thus we suppose the evaluation is fair to see each method's ability to address collider bias**.
>
> > **[W4] In the synthetic data in Table 1, some of the wins were not that big.**
>
> **Response:** We agree with you and **have adopted your kind suggestion in [W2.1]**. We think it mainly results from [W2.1] that the implementation of ShadowCatcher could be improved. We have adopted your suggestion to **use a random $Z$ instead of $-Z$ as the implementation of $Z^-$**. We re-run all experiments and achieve **obvious performance improvements.** Please kindly find the detailed results in the **response to [W2.1]**.
>
> > **[W5] There's a lot of experimental info missing from section 4 around what learning algorithm and models are used.**
>
> **Response:** We thank the reviewer for pointing out this issue. In the following, we **report the missing implementation details**, including what learning algorithms and models are used, as well as the choice of hyperparameters on different datasets in our revised version.
>
> Specifically, we adopt 3-layer neural networks to implement each module in ShadowEstimator and ShadowCatcher. We use the Adam optimizer with batch normalization in the training process, and we use the Wasserstein distance as the Integral Probability Metric (IPM) to implement all the methods that need IPM to balance representations. We implement all the methods in the PyTorch environment with Python 3.9. The CPU we use is 13th Gen Intel(R) Core(TM) i7-13700K, and the GPU we use is NVIDIA GeForce RTX 3080 with CUDA 12.1. **For the ease of reproducibility, the hyper-parameters of our methods on different datasets are detailed in the table below.**
>
> | Dataset | epochs | batch size | learning rate | weight decay | IPM weight | reject threshold |
> |:---------| :---------: | :---------: | :---------: | :---------: | :---------: | :---------: |
> |Synthetic datasets| 100 | 1024 | 0.03 | 0.01 | 0.001 | 1e-6 |
> | The IHDP dataset | 100 | 128 | 0.03 | 0.01 | 0.001 | 0.01 |
> | The Twins dataset| 100 | 1024 | 0.03 | 0.01 | 0.1 | 0.1 |
> | The Jobs dataset | 100 | 256 | 0.003 | 0.001 | 0.1 | 0.1 |
> | The ACIC 2016 datasets| 100 | 256 | 0.01 | 0.001 | 0.001 | 100 |
> | | | | | | | |

---

> > ### Author Response · Authors · 2023-11-21
> > **Responses by Authors (Part 3): More discussion on the limitations**
> >
> > > **[W6] End of Sec 3.3: it says "the final generated Z that passes the test" - should this be the first generated Z which passes the test?**
> >
> > **Response:** Thank you for the detailed comments. In fact, your understanding is right, and **the test can be regarded as a "stopping criterion".** The original statement "the final generated $Z$ that passes the test" means that **"after the first generated $Z$ passes the test, it can be used by ShadowEstimator to further estimate CATE under collider bias."** We apologize that our original statement does not seem to express such meanings, and we have revised it in the revised version on page 6.
> >
> > > **[Q1] What are the failure modes of this method - under what circumstances will learning a shadow variable be harder than others?**
> >
> > **Response:** Thank you for the comment. We believe **it depends on the extent to which covariates are involved in sample selection** in the data. If all covariates are highly involved in sample selection, it may be more difficult to extract representations that satisfy the assumption of shadow variables, that is, the conditional independence with $S$ given all covariates, treatment, and outcome.
> >
> > ***
> >
> > **We hope the above discussion will fully address your concerns about our work, and we would really appreciate it if you could be generous in raising your score.** We look forward to your insightful and constructive responses to further help us improve the quality of our work. Thank you!
> >
> > ***
> >
> > > **References**
> >
> > [1] Shalit, U., et al. Estimating individual treatment effect: generalization bounds and algorithms. ICML 2017.

---

> > > ### Comment · Reviewer_JJeK · 2023-11-22
> > > **Response**
> > >
> > > Thanks for the rebuttal - I found the clarifications helpful and I'm glad to see the suggestion improved your results! At the moment I will keep my score as is but will consider changing it in conversation with other reviewers.

---

> > > > ### Author Response · Authors · 2023-11-22
> > > > **Thank you!**
> > > >
> > > > We would like to thank the reviewer once again for the useful suggestions on our work. We look forward to your continued support of our work in the upcoming discussions. Thank you!

---

### Author Response · Authors · 2023-11-21
**General responses and manuscript revision summary**

Dear reviewers and AC,

We sincerely thank all reviewers and AC for their great effort and constructive comments on our manuscript.

As reviewers highlighted, we believe our paper tackles an interesting and important problem (Reviewer JJeK, Reviewer G11c, Reviewer 5YRg, Reviewer Y4SN), providing a novel and effective (Reviewer JJeK, Reviewer G11c, Reviewer 5YRg) method for CATE estimation under collider bias, validated with extensive experiments (Reviewer JJeK, Reviewer G11c, Reviewer 5YRg) with a clear presentation (Reviewer JJeK, Reviewer G11c, Reviewer 5YRg, Reviewer Y4SN).

Moreover, we thank the reviewers for pointing out some typos and unclear figures or texts in our manuscript (Reviewer JJeK, Reviewer G11c, Reviewer 5YRg), as well as for the suggestions for comparing the most recent CATE estimation baselines on more comprehensive benchmark and real-world datasets (Reviewer Y4SN). In response to these comments, we have carefully revised and enhanced our manuscript with the following additional discussions and extensive experiments:

- [Reviewer Y4SN] **We conducted experiments on two more datasets with more comprehensive DGPs,** namely **Jobs** and **ACIC 2016**, and **removed CATE evaluation results on Twins dataset** in Table 2.
- [Reviewer Y4SN] **We compared with more recent CATE estimation baselines** [1-5] to both synthetic and real-world experiments, **in which [5] is the state-of-the-art method published in NeurIPS 23,** and the results are added in Tables 1-3 and Table 5.
- [Reviewer JJeK] Following the suggestion of Reviewer JJeK, **we changed the choice of $Z^-$ from $-Z$ to a random $Z$ in the continuous setting** on page 6. **We also conducted experiments with this new implementation**, and marked as **Ours (New)** in the revised Tables. Impressively, compared to our original implementation marked as **Ours**, such change results in a significant performance improvement. Thank you for such useful suggestions.
- [Reviewer JJeK] **We clarified with more implementation details**, including what learning algorithms and models are used and the choice of hyper-parameters on different datasets in Table 4 and Appendix A.2.1 on Page 14.
- [Reviewer JJeK] **We added a detailed explanation of Equation (5)** on page 18.
- [Reviewer 5YRg, Reviewer JJeK] **We corrected the typo of $L_q$ on Page 6**, with a detailed explanation on page 18.
- [Reviewer G11c, Reviewer 5YRg] **We adjusted Figures 2 and 3** on pages 5 and 7 to make them clearer.

These updates are temporarily highlighted in $\textcolor{red}{red}$ for facilitating checking. We hope our response and revision can address all the reviewers' concerns, and we are more than eager to have further discussions with the reviewers in response to these revisions.

Thanks,

Submission5512 Authors.

***

> **References**

[1] Greiner, N. H. R. Learning Disentangled Representations for CounterFactual Regression. ICLR 2020.

[2] Zhang, W., et al. Treatment Effect Estimation with Disentangled Latent Factors. AAAI 2021.

[3] Zhong, K., et al. DESCN: Deep Entire Space Cross Networks for Individual Treatment Effect Estimation. KDD 2022.

[4] Wu, A., et al. Learning Decomposed Representations for Treatment Effect Estimation. TKDE 2022.

[5] Hao, W., et al. Optimal Transport for Treatment Effect Estimation. NeurIPS 2023.

---

### Meta-Review · Program_Chairs · 2024-01-31

**Metareview:**

This paper explores the question of how to handle colliders (where conditioning on the variable induces dependence between covariates). The work leverages classical identifiabiliy results that show that effects are identifiable in this scenario in the presence of shadow variables (which satisfy two key conditional independence criteria). To find shadow variables this work uses a two step approach - first shadow variable representations are generated and then a test is used to ensure the conditions for a valid set of shadow variables are met.

The core idea behind the work is sensible as a collection of three ideas wrapped in one helping use ML towards improving causal inference. The reviewers found the paper good, if at times loose with notation and adhoc in the collection of methods it uses with little to no effort spent thinking about theoretical guarantees about the resulting method. At the end of the discussion period, the authors replaced the experiments on the Twins study with the Jobs and ACIC data and incorporated another baseline into the paper. While these changes are welcome and serve as improvements to the manuscript, there still remain atleast two avenues for improvement from a technical perspective.

First, as reviewer JJeK points out, there was little discussion on the limitations and failure modes of the method. While this was partially improved in the rebuttal, I think there is still room for improvement. For example, what happens if no valid shadow variables are found -- what is the output of the method when there are many weak shadow variables vs one/two strong ones (strength in terms of the link function with respect to the outcome variable). These ablations and experiments are *particularly* important for a paper that claims to have no natural baselines to compare against since subsequent methods will compare against this one.

Second, In the absence of theory to explain the validity of the method, there is much more need for empirical rigor to ensure practitioners know when such a method is safe to use. Finally, as suggested by Y4SN, I think a broader swathe of experimental results the result of the method under different generative processes (whose conclusions are highlighted in the main paper) showcasing when the method does (and does not) work would be warranted.

PC/SAC comment: After calibration and downweighting inflated and non-informative reviews, this paper is assessed to fall below the accept threshold.

**Justification For Why Not Higher Score:**

See meta-review

**Justification For Why Not Lower Score:**

N/A

---

### Decision · Program_Chairs · 2024-01-16

Reject